# Research Based on Modeling and Simulation of the Transient Regime in Controlled Switching with High Power Switches

**Caius Panoiu, Dumitru Ciulica, Manuela Panoiu * and Sergiu Mezinescu**

Department of Electrical Engineering and Industrial Informatics, University Polytechnica Timisoara Hunedoara, 331128 Hunedoara, Romania; caius.panoiu@fih.upt.ro (C.P.); dumitru.ciulica@gmail.com (D.C.); sergiu.mezinescu@gmail.com (S.M.)
\* Correspondence: manuela.panoiu@fih.upt.ro; Tel.: +40-741012314

**Abstract:** This paper addresses one of the current areas of interest in electrical engineering, which is controlled switching of high voltage circuit breakers. During their operation, the problem of controlled switching of high voltage circuit breakers in commutation regimes was studied. Several types of switching were analyzed, considered representative of the transient regime, depending on the type of load, on the defect that may occur on the power supply lines, as well as depending on the position of this defect (near or far). The study carried out in the paper includes simulations of the controlled connection/disconnection operations in a transient regime, assuming the existence of different kinds of defects. To perform the study and simulations in the transient regime, a model, implemented in Matlab, was used for a time interval located around the origin of the time axis. The study included the dependence of the SF6 circuit breaker switching process on the following parameters: the DC voltage supply, ambient temperature and oil pressure in the circuit breaker actuator. The validity of the theory presented in this paper, in addition to being validated by simulations, is proven by the fact that the protection system currently in use at the power station of an 800 MW power plant, at the 400 kV power line, is based on the principles presented in this paper. The theory presented in the paper has been implemented in industry for nearly two years, and the results confirm that the theory presented in the paper is fully applicable in high voltage power stations.

**Keywords:** circuit breaker; connection and disconnection operation; transition; modelling





## 1. Introduction

Given the growing role of electricity use in modern society, an interruption of power supply is an important problem that must be avoided. This is where the role and functions of circuit breakers come into play, which must ensure a quality electrical power supply. Circuit breakers are very important circuits in the power supply of electrical systems. They can handle faults that, if not properly managed, can cause power supply system failures.

This paper is an extended version of the conference paper [1] presented at the 2020 8th International Conference on Control, Mechatronics and Automation (ICCMA). In the conference paper, the main aspects are related to the disconnection of circuit breakers.

In this paper, we further study the switch connection regime of the circuit breakers, a regime that was not studied in the paper presented at the conference. As a result of the further study regarding the switch connection regime of the circuit breakers, the „Simulation in Matlab of the Transient Regime of the SF6 Circuit-Breaker Connection" chapter of this paper was developed. Theoretical elements were detailed with mathematical equations that show why the transient regime occurs in the case of connection and disconnection processes. Thus, in addition to aspects presented in [1], referring to the disconnection of circuit breakers, the dependence of the connection/disconnection process on certain parameters that influence these processes has been studied in this paper.

## 2. Related Works

Switching operations are performed not only to isolate faults, but also to connect or disconnect parts of the power supply network to achieve the purpose of isolating parts of the system, for maintenance or balancing production and consumption. Any switching operation determines a change in the state of the system. Because an ideal switch between two zones is not physically possible, a transient response within the system will inevitably occur during this switching process, which can determine the appearance of overcurrent or overvoltage. For each type of load, comparative analysis was performed to assess the influence of the parameters on the switching process, such as: ambient temperature, the supply voltage of the triggering coils, as well as the pressure of the hydraulic agent of the actuator.

The paper presents, in detail, the controlled switching process of high voltage circuit breakers (HVCB) and describes the theory and technology of the circuit breakers that are used today [2,3]. The paper discusses the benefits of controlled switching as well as how to determine the optimal time command of the controlled switches in order to reduce the effects of the transient switching regime. The study presented in this paper also refers to the transient regimes that occur. The research presented in this paper is based on [2,4,5], which detail the HVCB controlled switching.

The simulations performed confirm the benefits of controlled switching, namely improvements in circuit breaker performance, reduction in switching in the transient regime, reduced equipment maintenance costs, extended equipment life and improvements in energy quality. These advantages are also presented in [2].

The process of interruption of high voltage circuits has been studied in the literature in a large number of scientific papers, because electricity is widely used in all fields of activity. However, the need for electric power quality has increased more and more [2]. Customers expect that electric power to be supplied without interruption and at the best values of electrical parameters. Under these conditions, the disturbance levels are not acceptable, because they can lead to a series of failures [2,3].

Gas Insulated Substations (GIS) are currently widely used due to their operational safety [6]. In such substations, high voltage circuit breakers are widely used because they can quickly interrupt the fault and, thus, can prevent damage to electrical equipment [6]. Based on an HVCB study, the authors of [7] analyzed the defects that appear during switching of these types of GIS, as well as the failure of GIS isolation in the switch opening and closing process.

In [8], a close–open time control method is proposed using an auxiliary switch. According to this paper, the average close–open time for SF6 breakers was determined statistically to be 23–35 ms.

In [9], research is performed on transient circulation current that appears in the switching process of SF6 circuit breakers. A method for accurately calculating the transient circulation current is presented here.

The process of dynamic disconnection of electrical circuits takes place simultaneously with the ignition between the contacts of the switching equipment, which are called electrodes, through the column in which the current continues to flow [10]. The final extinction of the electric arc occurs at the zero crossing of the current, compared to which the transient recovery voltage has a sufficiently low growth rate, so that it can no longer produce the reigniting of the electric arc. Different mathematical models of circuit breakers have been developed as tools to assess circuit breaker capacity and to investigate the interaction between circuit breakers and external circuits.

In [11], the breaking capacity of a circuit breaker is analyzed, both for high currents and also for relatively small capacitive and inductive currents. In the case of inductive currents, a large amount of energy is stored in the inductive elements, which can generate an overvoltage when the current breaks. When the capacitive currents are interrupted, there is a danger of unsuccessful interruption at the first current crossing through zero.

In the cases of both capacitive and inductive switching, the process is influenced by the transient recovery voltage (TRV).

The authors of [12] investigate the measurement of transient pressures in a high voltage circuit breaker of 252 kV during high current interruption.

In [13] the impact of electric arc ignition speed of SF6 circuit breakers on arc contact erosion is analyzed. The study proved that the electric arc ignition speed has a significant effect on contact erosion.

In [14], the self-powered ring main unit is studied. Arc quenching methods are also studied to make this analysis possible. A main ring unit is used to perform voltage and current measurements of the power supply grid in order to detect failures.

### 3. The Switching Process

The current interruption of a circuit breaker normally occurs at zero value of current, in a time interval of microseconds. In the interruption process of the current, several processes take place simultaneously [15].

After the current is interrupted, the arc voltage remains constant in the high current range, rises to a peak value known as the extinction voltage, and then falls to zero with a very steep slope of the variation du/dt.

The current crosses the zero value with a variable slope of di/dt, but can be distorted under the influence of the arc voltage. The electric arc has a resistive character and, therefore, the arc voltage and current reach zero value at the same time, as demonstrated in [16,17].

Around the zero value of the current, the energy in the arc channel is quite low (practically at the zero value of the current, and there is no increase in energy) and when the arc length is at its maximum, the current can be interrupted.

After the power supply is interrupted, the gas between the switch contacts is still hot. The process is characterized by a steep slope of the recovery voltage and of the resulting electric field. The charged particles begin to circulate and cause a current called the post-arc current. The presence of the post-arc current, combined with the transient recovery voltage, conducts energy consumption in the hot gas channel.

At the beginning of the extinguishing process, when the energy of the arc is formed by the individual gas molecules dissociated into free electrons and positive ions, the plasma state of the arc is restored, and the current cannot be interrupted. This phenomenon is called thermal switching of the circuit breaker.

When the power supply disconnect is achieved, the hot gas cools and the post-arc current disappears. Another electrical discharge can occur when the dielectric strength of the space between the circuit breaker contacts is not sufficient to withstand the transient recovery voltage.

In [18], the current interruption process of a circuit breaker is presented. It was shown in [18] that there is a strong interaction between the physical process involving the breaker contacts and the network connected with the terminal of the breaker. This leads to the simplest lumped-element representation of the power system. Figure 1 presents the elements of the electrical network connected to the circuit-breaker terminals. All the variables used in this research paper are listed in the nomenclature section.

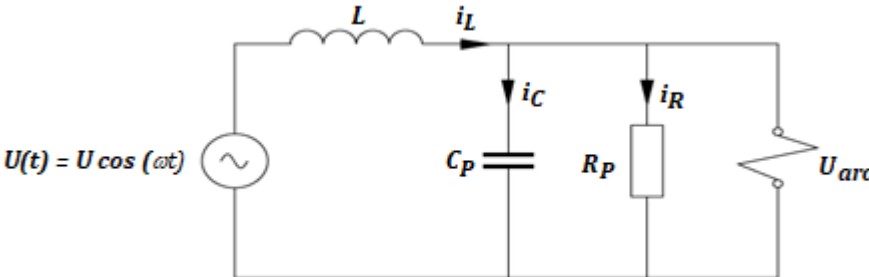

**Figure 1.** Electrical network elements connected to the circuit-breaker terminals.

The current is canceled by a combined action of $R_p$ and $C_p$, the arc energy decreases, and the cooling of the mechanism has a period which depends on the time constant of the circuit. In [18], it was demonstrated that the equation of the interruption time is presented in (1), where $u_0$ represents the constant arc voltage, which causes a resistive current [18].

$$\Delta t = \Delta t_R + \Delta t_C = \frac{u_0/R_p + C_p\left(\frac{du_{arc}}{dt}\right)}{di_L/dt} \tag{1}$$

When the current is interrupted, the voltage on $C_p$, and on $R_p$, is equal to zero. The transient recovery, at the appearance of the voltage between the circuit breaker contacts, presumes the charging of the $C_p$ capacity and causes the so-called time delay of the waveform, TRV [19].

To increase the breaking capacity of the circuit breaker, a parallel capacitor is mounted on the SF6 circuit breakers. After a disconnecting operation, the transient currents will circulate through the system and, after a connecting operation, when a high frequency current is interrupted, a transient recovery voltage or TRV will appear at the circuit-breaker terminals [19].

The configuration of the electrical network at the circuit-breaker terminals determines the amplitude, frequency and shape of the current and voltage oscillations.

When capacitor banks used for voltage control are placed in a station, the disconnecting devices interrupt a mainly capacitive load when operating under normal charging conditions. The phase difference between current and voltage is about 90° on a capacitive charge.

When a high power transformer is disconnected in a normal load situation, the current and voltage are out of phase by about 90° on an inductive charge.

A fault usually consists of a short circuit. Figure 2 depicts a fault between one phase and null on a high voltage transmission line at distances ranging from a few hundred meters to a few kilometers from the circuit-breaker terminals.

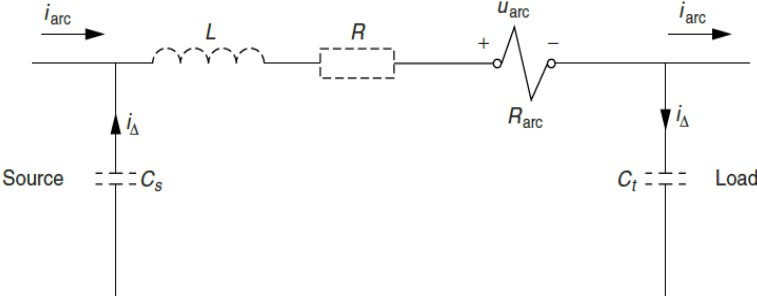

**Figure 2.** Interruption of the small inductive current.

In [18], the difference between current chopping and virtual chopping was presented, and it was specified that in the case of virtual chopping, the arc is made unstable through a superimposed high-frequency current caused by oscillations with the neighboring phases in which current chopping took place. Virtual chopping has been observed for gaseous arcs in air, SF6 and in oil. Based on this observation, Figure 2 shows the lumped-element representation of a breaker interrupting a small inductive current. The relationship between arc voltage and current is presented in Equation (2).

$$(u_{arc}i_{arc})^\alpha = \eta = ct, \tag{2}$$

The parameter $\alpha$ is a constant in the range (0.4 and 1) [18]. If the high frequency oscillation causes a small disturbance of the arc current, and assuming that the arc channel is in steady-state, the Equation (2) determines the relationship between the arc current and the arc voltage [20].

From Figure 2, Equation (3) can be deduced:

$$L\frac{d(i_{arc} + i_\Delta)}{dt} + R(i_{arc} + i_\Delta) + R_{arc}(i_{arc} + i_\Delta) + \int \frac{i_\Delta}{C}dt = u_0, \tag{3}$$

In [18], it was demonstrated that if $R_{arc}$ is steady-state arc resistance, the following can be derived from Equation (2):

$$R_{arc} = \frac{u_{arc}}{i_{arc}} = \frac{\eta}{(i_{arc} + i_\Delta)^{\alpha+1}}, \tag{4}$$

In Equation (3) $u_0$ is residual voltage on $C_S$ and $C_t$ and equivalent capacity $C = \frac{C_S C_l}{(C_S + C_l)}$.

By differentiating the Equation (3) and replacement of $R_{arc}$ and $dR_{arc}/dt$, the following is obtained:

$$L\frac{d^2 i_\Delta}{dt^2} + (R - \alpha R_{arc})\frac{di_\Delta}{dt} + \frac{i_\Delta}{C} = 0 \tag{5}$$

The characteristic Equation of (5) is:

$$\lambda^2 + \frac{(R - \alpha R_{arc})}{L}\lambda + \frac{1}{LC} = 0 \tag{6}$$

Solutions of Equation (6) are given in Equation (7):

$$\lambda_{1,2} = -\frac{(R - \alpha R_{arc})}{2L} \pm \sqrt{\left(\frac{R - \alpha R_{arc}}{2L}\right)^2 - \frac{1}{LC}} \tag{7}$$

According to Equation (7), the circuit from Figure 2 oscillates when the expression $\left(\frac{R - \alpha R_{arc}}{2L}\right)^2 - \frac{1}{LC}$ is negative as in Equation (8), i.e., [21]:

$$\left(\frac{R(R - \alpha R_{arc})}{2L}\right)^2 < \frac{1}{LC} \tag{8}$$

From the point of view of the values of $R$ and $\propto R_{arc}$, we can have three cases:

- If $\propto R_{arc} > R$ results in oscillations in which the amplitude increases, the system is unstable;
- If $\propto R_{arc} < R$, it is clear that the created regime is a depreciated regime;
- If $\propto R_{arc} = R$, a permanent oscillatory regime will result.

The value of the current depends on the extinguishing medium and the capacity of C, which is around 0.01–0.05 µF.

For gas extinguishing circuit breakers, the extinguishing current may vary from amperes for SF6 up to tenths of amperes for vacuum circuit breakers.

Controlled switching, also known as point-on-wave (POW) switching, is defined as control operations of the switch so that in each phase of an AC network power supply, the current interruption occurs at moment times that are optimal for the switch and switched load, and does not affect power quality, as shown in [1,22]. Point on wave prevents high inrush currents that are associated with the mechanical switching of transformers, reactors and capacitors, as demonstrated in [22–24].

The electric arc, as a circuit element, is characterized by a nonlinear dependence between its voltage and current and also by a resistive character, as detailed in [1,25,26]. Figure 3a represents the electric scheme of the circuit, illustrating the interruption of a short-circuit current. It contains an $I_1$-main switch and an $I_2$-auxiliary switch. Switching the $I_2$ to ON, it is possible to discharge the capacitor Co, which is initially loaded with the polarity from the figure. At the interruption of a short-circuit current $i_c(t)$ simultaneously

with the switching $I_1$ to OFF, the switch $I_2$ is turned to OFF, so the intensity i(t) a of the current through the arc results from Equation (9).

$$i(t) = i_k(t) - i_c(t), \tag{9}$$

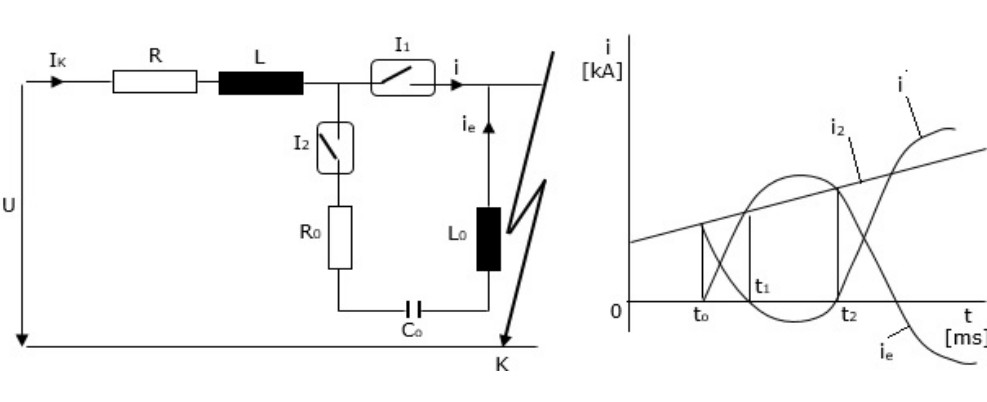

(**a**)                                                   (**b**)

**Figure 3.** Modelling of the electric arc: (**a**) electric scheme of the circuit; (**b**) current variation [1].

The arc extinction process involves disconnected circuit parameters (voltage transient recovery, which produces a request for dielectric current interruption, which is conducive to circuit breaker heating) as well as the circuit breaker-specific parameters (breakdown voltage chamber extinction, which expresses the speed of dielectric rigidity restoration, and also the electric arc voltage, which is dependent on the degree of cooling and the extinguishing medium).

In short intervals, containing the moments of current cancellation, the arc column temperature, and also its conductance, are rapidly decreased, demonstrating a process of recovery of the dielectric strength, as shown in [22,27–29].

When the current is cancelled, corresponding to the periodic extinction of the arc current, the transient voltage is applied between the contacts of the circuit breaker. In general, it can be assumed that the definitive extinction of the electric arc is obtained at the moment of the normal cancellation of the current, according to which the transient voltage has a growth rate sufficiently low that it cannot produce the rebound of the electric arc, neither by thermal packing nor by thermodynamic breakthrough. These considerations refer to the usual cases of dynamic disconnection in alternating current installations, corresponding to either normal load or short-circuit regimes.

## 4. Materials and Methods

### 4.1. Simulation In Matlab of the Transient Regime of the SF6 Circuit-Breaker Commutation

A mathematical model was developed in Matlab for the time interval t = −100 ms and t = 200 ms using voltage and current Equations (10) and (11). Figure 4 depicts the mathematical model, which was also described in [1].

$$v(t) = \sqrt{2} \ V \sin(2\pi ft + \phi V), \tag{10}$$

$$i(t) = \sqrt{2} \ I \sin(2\pi ft + \phi I), \tag{11}$$

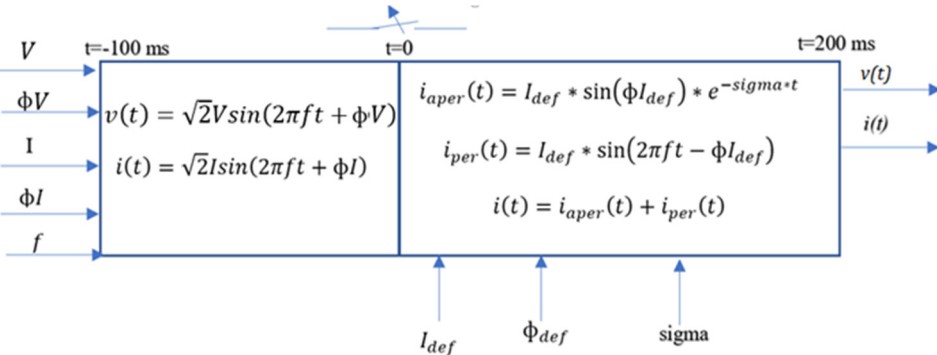

**Figure 4.** The mathematical model used [1].

The following parameters: voltage V [V], current I [A], frequency f [Hz], initial phase current ϕI and initial phase voltage ϕV are used for simulation for the time interval in the range of t ϵ (−100, 0) ms. Depending on the initial phase current, ϕI, an inductive, resistive or capacitive character results in the relations as in Equation (12).

$$\phi I \in [-\pi/2, 0), \text{ capacitive character}$$
$$\phi I = 0, \text{ resistive character}, \tag{12}$$
$$\phi I \in (0, \pi/2], \text{ inductive character}$$

The transient mode frequently occurs when the circuit breaker is opened due to a fault in the high voltage power supply lines. In this case, the transient current consists of two components: a periodic one, with frequency equal to the source frequency, which feeds the defect, and an aperiodic one, which is exponential and depends on the place of the defect on the line.

Using the Matlab simulation program, the time dependence of voltage and currents was obtained as presented in Figure 5. Figure 5 shows that the voltage phase and current phase are typical of the normal operation of a three-phase system in a time interval between −100 ms and 0 ms [29].

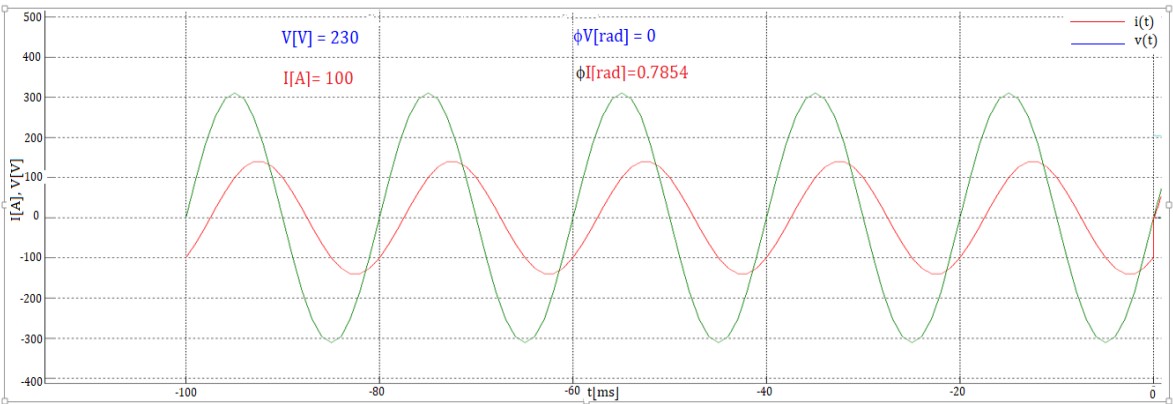

**Figure 5.** Voltage and current variation for the time interval t∈(−100,0) ms.

The program developed in Matlab aims to connect/disconnect a switch with the SF6 extinguishing medium in the case of a transient regime. To do this, the ideal disconnection/connection time pattern is used, as in Figure 6, at disconnection, and in Figure 7 during the connection process.

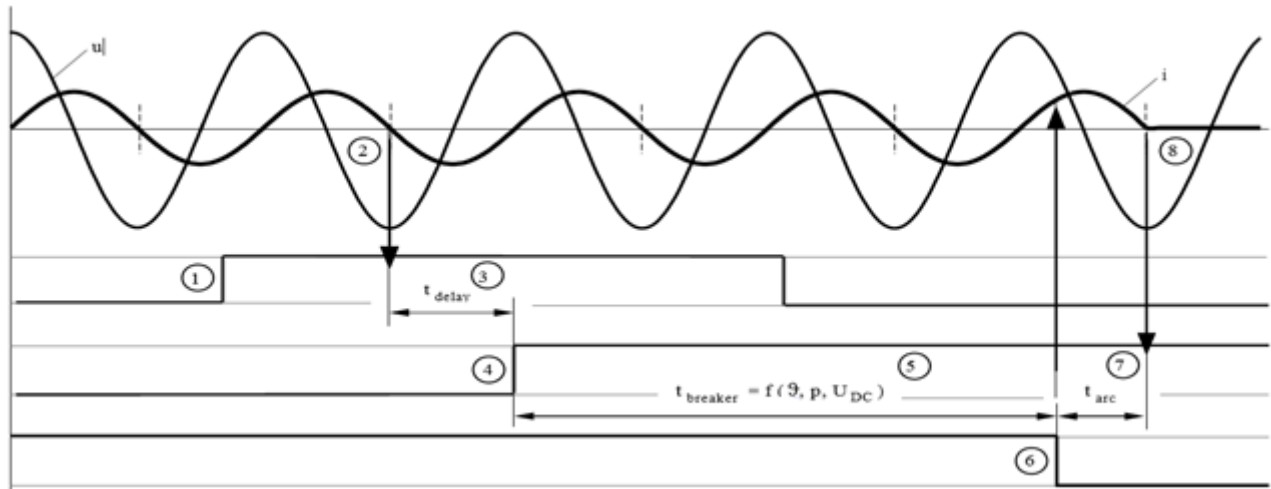

**Figure 6.** Disconnecting operation of circuit breaker with a control system: 1—disconnected command, 2—identify the zero point of the current, 3—delay time, 4—command to the breaker disconnected coil, 5—disconnected time, 6—separate contacts; 7—arc time, 8—end of the current flow.

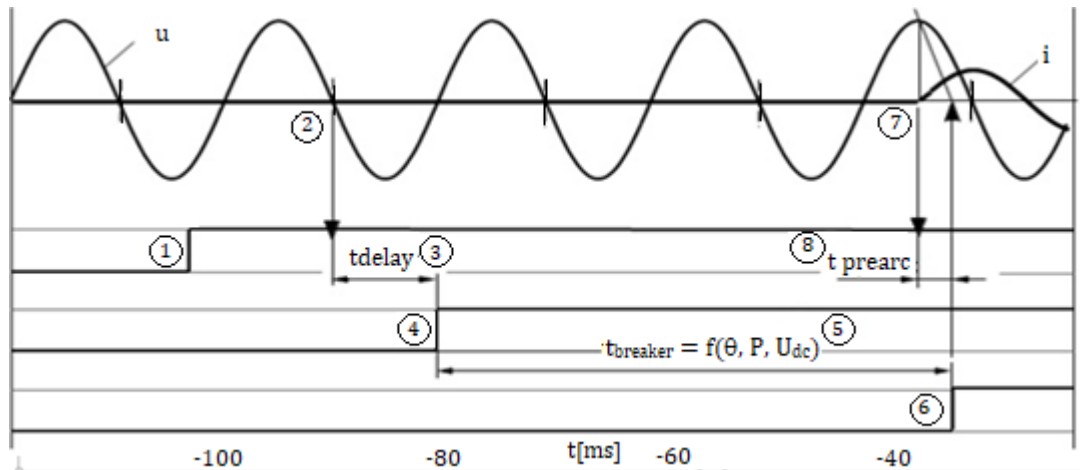

**Figure 7.** Connecting operation of circuit breaker with a control system: 1—connected command, 2—identify the zero point of the voltage, 3—delay time, 4—command to the breaker connected coil, 5—connected time, 6—touch contacts; 7—beginning of current flow, 8—prearc time.

When the circuit breaker has variations in external conditions during operation, corrections are required depending on their external conditions. When the switch breaker has time variations of the external conditions during operation, corrections are required depending on these variations. The checking of control of the variations of the mechanical quantities is performed according to the variations of the environmental conditions, such as the influence of the idling time, variation of the temperature, variation of the coil voltage, the change of the hydraulic agent pressure and the changing in SF6 gas pressure.

Variations in ambient temperature can be compensated for using appropriate transducers. The need for compensation depends on the size of the variation and the actual operating conditions. For proper operation, adaptive control can be good enough if sufficiently small variations in the adjustment process are taken into account. The system estimates the expected operating time of the circuit-breaker for variations in external temperature, pressure, SF6 gas pressure and for the auxiliary voltage supply. The most important factors that influence the operating time of the circuit-breaker are: the control voltage on the CONNECT/DISCONNECT coil, the ambient temperature where the circuit-

breaker is located, as well as the hydraulic pressure for circuit-breakers with hydraulic actuation of mechanisms.

Table 1 shows the nominal values and the limit values of the studied circuit-breaker parameters [30].

**Table 1.** Usual parameter values of the studied circuit-breakers.

| Parameters | Min. | Typ. | Max. |
|---|---|---|---|
| Frequency [Hz] | 48 | 50 | 52 |
| Voltage control [V] | 187 | 242 | 255 |
| Voltage connect control [V] | 187 | 242 | 255 |
| Voltage disconnect control [V] | 154 | 241 | 255 |
| Hydraulic pressure [bar] | 0 | 355 | 400 |

4.1.1. Simulation in Matlab of the Transient Regime of the SF6 Circuit-Breaker Disconnection

Considering the moment t = 0 as the moment of occurrence of the transient regime, we use the following mathematical Equations for currents (13), (14) and (15)

$$i_{aper}(t) = I_{def} \cdot \sin(\varphi I_{def}) e^{-\sigma t}, \tag{13}$$

$$i_{per}(t) = I_{def} \cdot \sin(2\pi f t - \varphi I_{def}), \tag{14}$$

$$i(t) = i_{aper}(t) + i_{per}(t), \tag{15}$$

In these Equations

- $i_{aper}$ is the aperiodic component of the current in the transient regime;
- $i_{per}$ is the periodic component of the current in the transient regime;
- $i(t)$ is the transient current, as the sum of the two components.

In Figure 8, the two components of the transient current are represented as well as their sum, which is actually the defective current.

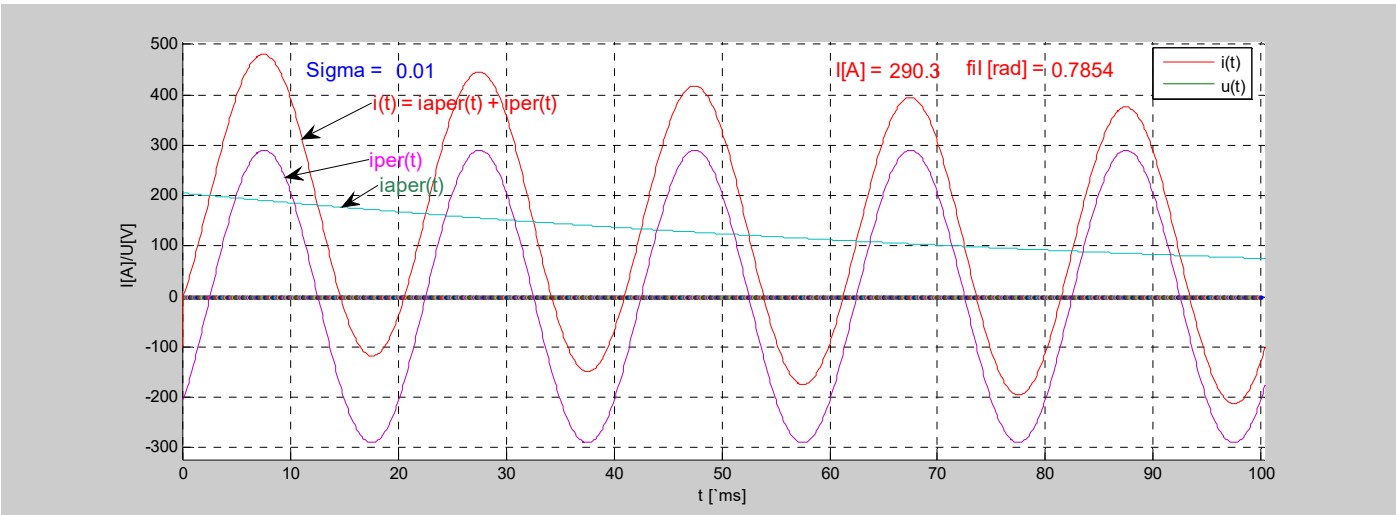

**Figure 8.** Components of the transient characteristic on inductive load; over time, t∈(0,100) ms.

For the transient regime, the phase $\varphi I_{def}$ of the current $I_{def}$ represents the phase shift of the source and determines the load character, like in Equation (12). Based on the Equations (13)–(15), the components of the transient characteristic t∈(0,100) ms were obtained over time, as shown in Figure 9, using an inductive load, and as shown in Figure 10 for a capacitive load. In Figure 9, the voltage and current characteristic of a transient regime are represented, where σ = 0.01, so the defect is at a long distance and the periodic component of the current is equal to $\pi/2$, the amplitude of the defective current is

I = 290.3 A. In this case, it is noted that the aperiodic component is above the horizontal axe where the transient current is found in the positive area.

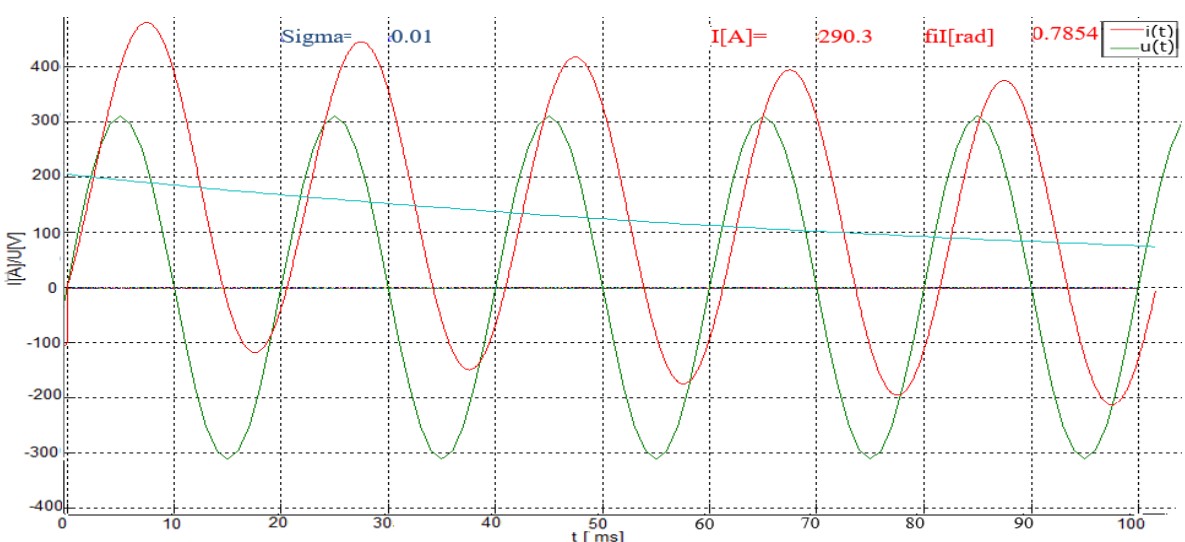

**Figure 9.** Components of the transient characteristic on capacitive load; over time, t∈(0,100) ms.

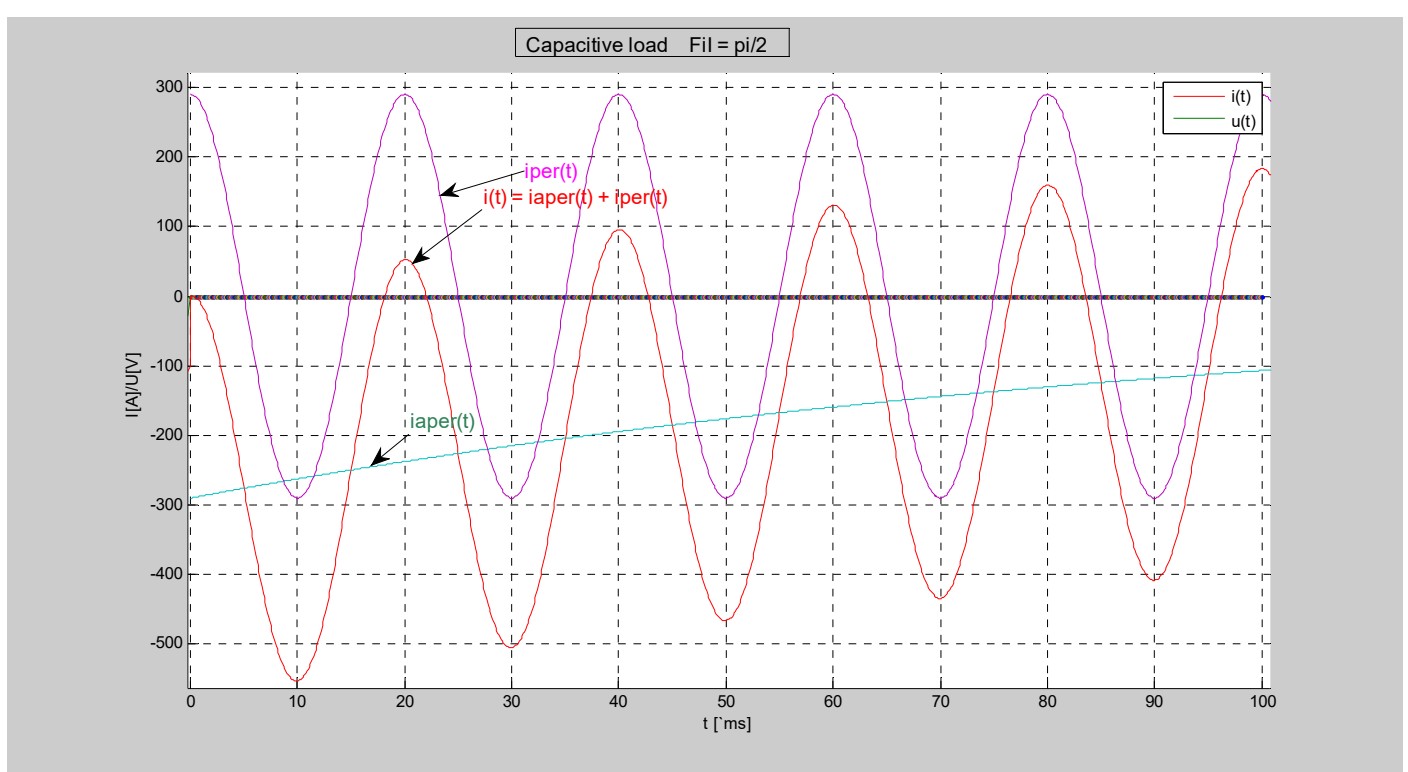

**Figure 10.** Characteristic transient current capacitive load.

If the fault phase is negative $\phi I_{def} = -\pi/2$, the single-phase characteristic of the transient current $I_{def}$ = 290.3 A in the time interval t∈(0,100) ms, for a long distance defect, σsigma = 0.01 as in Figure 10, and it is found that the maximum transient current is negative [1].

To study the influence of the fault (defect) location, the coefficient of σ = 0.9 is changed to 0.09, 0.009 and 0.0009 from the defect from near to long distance. In Figure 11, it is noted that flattening the aperiodic component is more pronounced for near defects, i.e., σ of

about 1. Figure 11 shows the voltage and current variation for inductive load, $\phi I_{def} = \pi/2$, $I_{def} = 290.3$ A; over time, $t \in (-100,100)$ ms, for $\sigma = 0.9$, $\sigma = 0.09$, $\sigma = 0.009$ and $\sigma = 0.0009$.

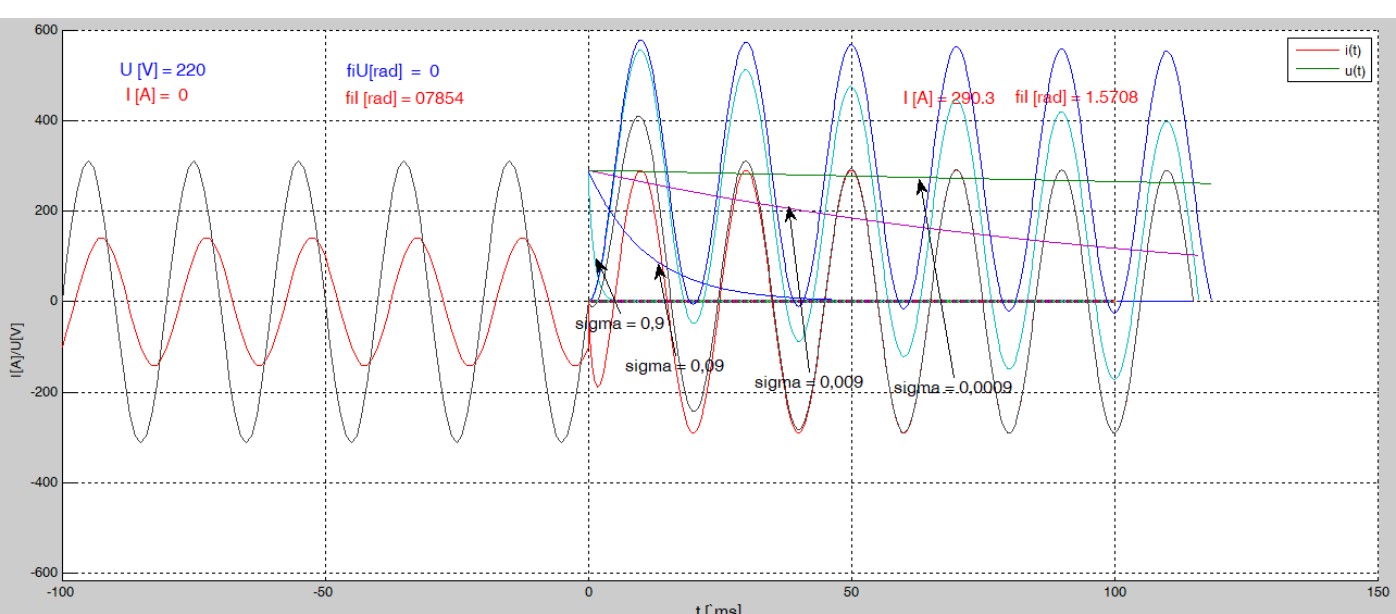

**Figure 11.** The voltage and current variation for inductive load, $\phi Idef = \pi/2$, $Idef = 290.3$ A; over time, $t \in (-100,100)$ ms for $\sigma = 0.9$, $\sigma = 0.09$, $\sigma = 0.009$ and $\sigma = 0.0009$.

It is observed in Figure 11 that when a transient regime appears, the current to be interrupted is very high, and if the moment of connecting the circuit breaker is in a maximum wave, this current affects the circuit breaker [1].

For simulation, a time control was used, which is represented in blue color. Depending on the type of circuit breaker, a time characteristic of each delay switch is calculated, which is dependent on the outdoor temperature, on the DC voltage supply of the switch trigger coil and on the oil pressure if the circuit breaker is operated hydraulically, i.e., $t_{breaker} = f$ (°C, V_DC, p), as shown in [31–33]. The calculation of the coefficient according to which the disconnection of the switch is studied is conducted based on measured values and saved in vectors. The coefficients will then be calculated by interpolation.

In the scope of studying the disconnection of a circuit breaker, the scheme in Figure 4 was used. The time coefficients on the disconnection process of the circuit-breaker were experimentally determined for the studied circuit-breakers and the results are presented in the following.

1.  The temperature dependency of time coefficient $C_{temp} = f(T)$ on the disconnection of a circuit breaker is represented in Table 2 [34]. Taking into consideration the measured values, the equation which describes this dependency is presented in Equation (16).

**Table 2.** Measured values of the dependency of time coefficient on outdoor temperature.

| T [°C] | $C_{temp}$ (ms) |
|---|---|
| 40 | 21.5 |
| 20 | 22 |
| 0 | 22.5 |
| −20 | 23 |
| −30 | 24 |
| −40 | 31 |

$$C_{temp}(T) = 0.0026 \cdot T^2 - 0.0829 \cdot T + 21.482 \tag{16}$$

Figure 12 represents the dependency of parameter $C_{temp}$ on the outdoor temperature on the disconnection of a circuit breaker.

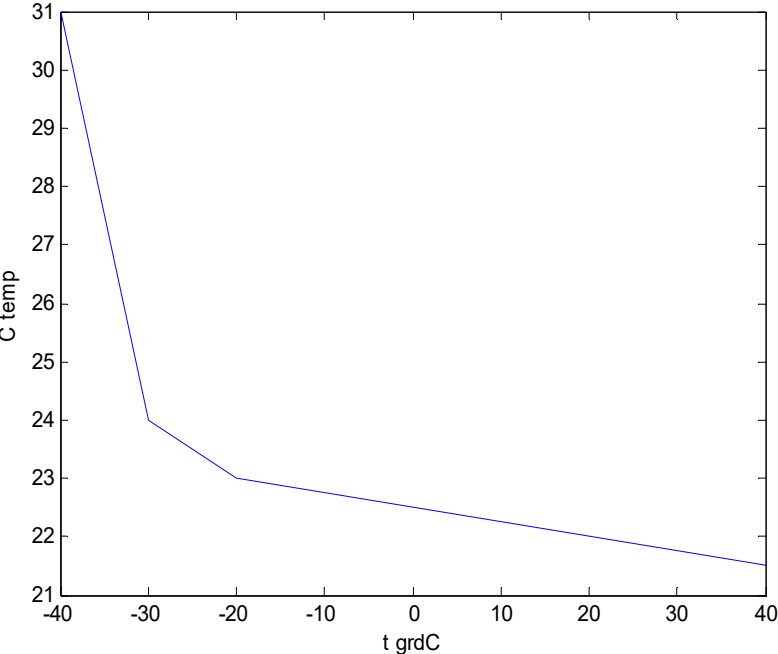

**Figure 12.** Dependency of $C_{temp}$ on the outdoor temperature on the disconnection of a circuit breaker.

2. The DC voltage supply dependency of time coefficient $t_{UDC} = f(UDC)$ on the disconnection of a circuit breaker is represented in Table 3 [34]. Taking into consideration the measured values, the equation which describes the dependency is presented in Equation (17).

**Table 3.** Measured values of the dependency of time coefficient on DC voltage control.

| U Control (V) | $t_{UDC}$ (ms) |
|---|---|
| UDC | |
| 187 | 6.6 |
| 192 | 5.6 |
| 203 | 3.4 |
| 209 | 2.2 |
| 221 | −0.2 |
| 232 | −2.4 |
| 241 | −4.2 |
| 252 | −6.4 |
| 255 | −7 |

$$t_{UDC}(UDC) = -0.2 \cdot UDC + 44 \tag{17}$$

Figure 13 represents the dependency of parameter $t_{UDC}$ on the DC voltage supply on the disconnection of a circuit breaker.

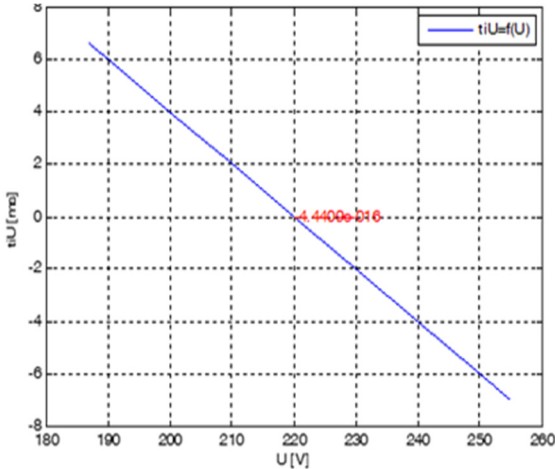

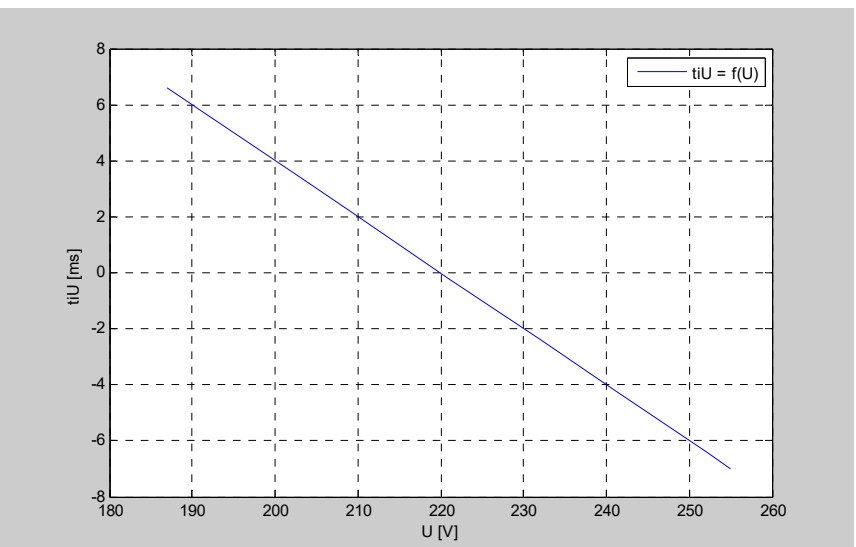

**Figure 13.** Dependency of $t_{UDC}$ on the DC voltage supply on the disconnection of a circuit breaker.

3.　The oil pressure dependency of time coefficient $C_{Ph}$ = f(P) on the disconnection of a circuit breaker is represented in Table 4 [34]. The equation that describes the dependence is presented in Equation (18), taking into account the measured values.

**Table 4.** Measured values of the dependency of time coefficient on oil pressure.

| Oil Pressure (bar) | $C_{Ph}$ (ms) |
|---|---|
| 310 | 6 |
| 316 | 4.2 |
| 321 | 2.7 |
| 326 | 1.2 |
| 333 | −0.9 |
| 337 | −2.1 |
| 342 | −3.6 |
| 346 | −4.8 |
| 350 | −6 |

$$C_{Ph}\,(P) = -0.3\,P + 99 \tag{18}$$

Figure 14 represents the dependency of parameter $C_{Ph}$ on the oil pressure on the disconnection of a circuit breaker.

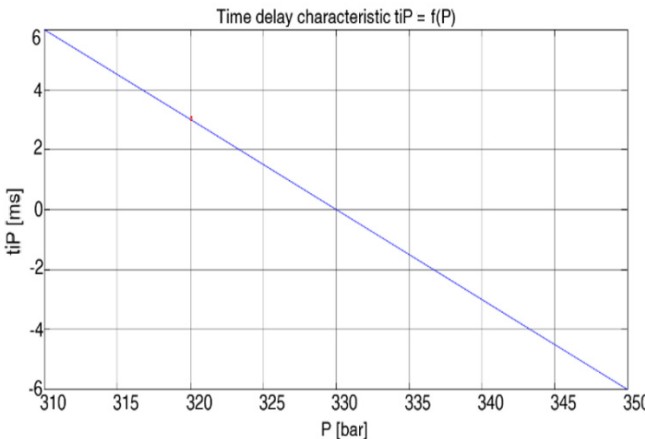

**Figure 14.** Time delay characteristic of the oil pressure on the disconnection of a circuit breaker.

Calculation of the final $t_{breaker}$ parameter is made according to Equation (19) [35].

The program, implemented in Matlab, commands the disconnection of the circuit breaker. The assured time command precision is 1 microsecond and depends on the outdoor temperature, on the DC voltage supply of the switch trigger coil and on the oil pressure. Based on this precision, the program determines the next zero crossing where the switch is disconnected.

$$t_{breaker} = C_{temp} + t_{UDC} + C_{Ph}, \tag{19}$$

For simulation of the disconnecting process, the following parameter values of a monophasic power supply system were used: U = 8.6 V, ɸU = −0.34°, f = 50 Hz, I = 3 A, ɸI = 1.26°, outdoor temperature = 23 °C, Ucc = 242 V, P = 330 bar. In Figure 15, the simulation results for the time command of disconnected of circuit breaker of $t_1 = t_{command} = 13$ ms are presented.

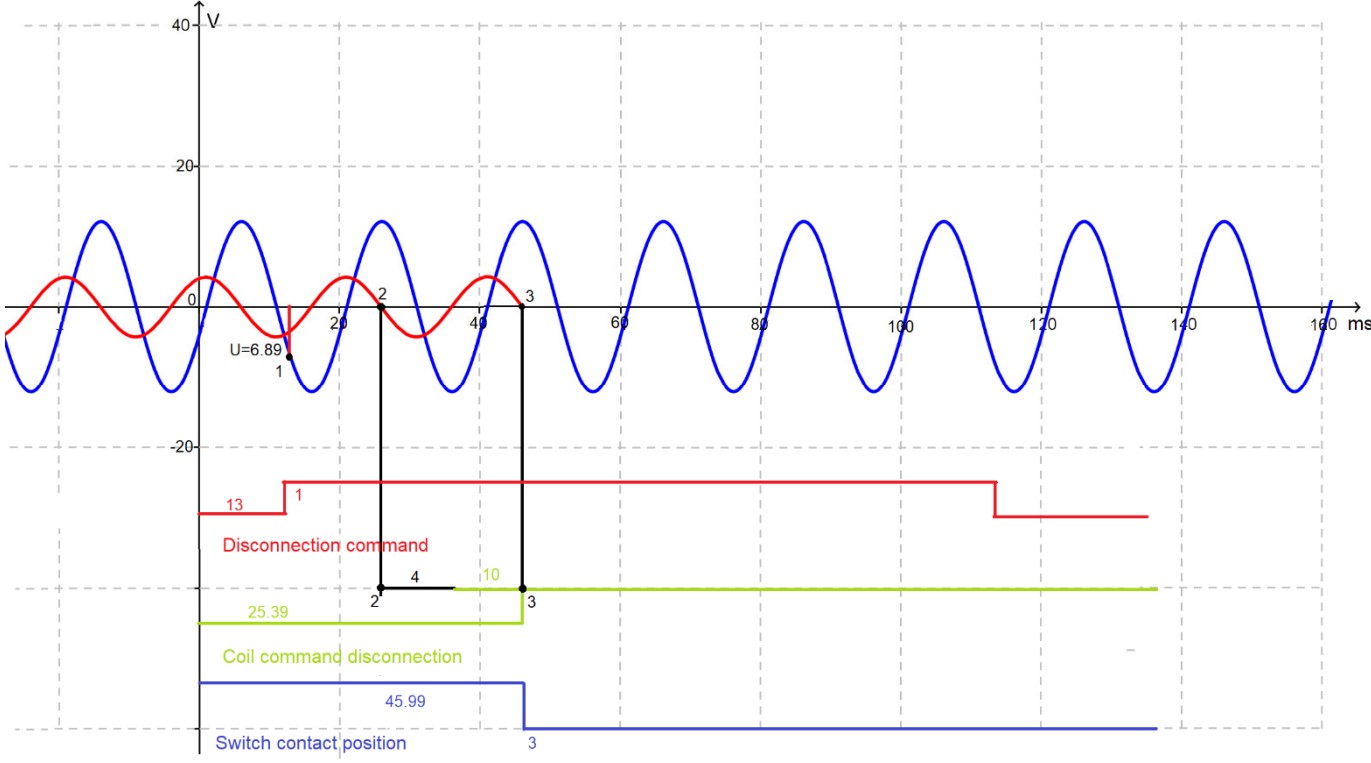

**Figure 15.** The simulation of monophasic power supply system on the disconnecting process of the circuit breaker on $t_1 = t_{command} = 13$ ms.

The main parameters which are used in simulation of the disconnection process of the circuit-breaker are presented in Table 5. Based on the equation presented in Table 6, the parameter values of the disconnection simulation process are $t_1 = t_{command} = 13$ ms.

**Table 5.** The main parameters used in simulation of disconnection process of the circuit-breaker.

| Parameters | Values |
|---|---|
| Temperature [°C] | 23 |
| Ucc [V] | 242 |
| P [bar] | 330 |
| tT | 21 |
| tucc | −4 |
| tp | 0 |
| t5 | 17 |
| t1 | 13 |
| t2 | 26 |

**Table 6.** The equation used in simulation of the disconnection process of the circuit-breaker.

| Equation | Parameters | Values |
|---|---|---|
| i(t) = 0 | t2 | 26 |
| t = f(T,U,P) | t5 | 17 |
| SP2 = 1000/f/3 | sp2 | 10 |
| nrp = t5/SP2 + 1 | Nrp | 2 |
| t7 = t2 + nrp*SP2 | t7 | 46 |
| t4 = t7 − t6 | t4 | 29 |
| t3 = t4 − t2 | t3 | 3 |

Using a program which was developed in Matlab, the need to control the circuit breakers when the current crosses zero value, in the case of a transient current, was demonstrated.

Figures 16 and 17 present the disconnecting operation of the circuit breaker for an inductive load.

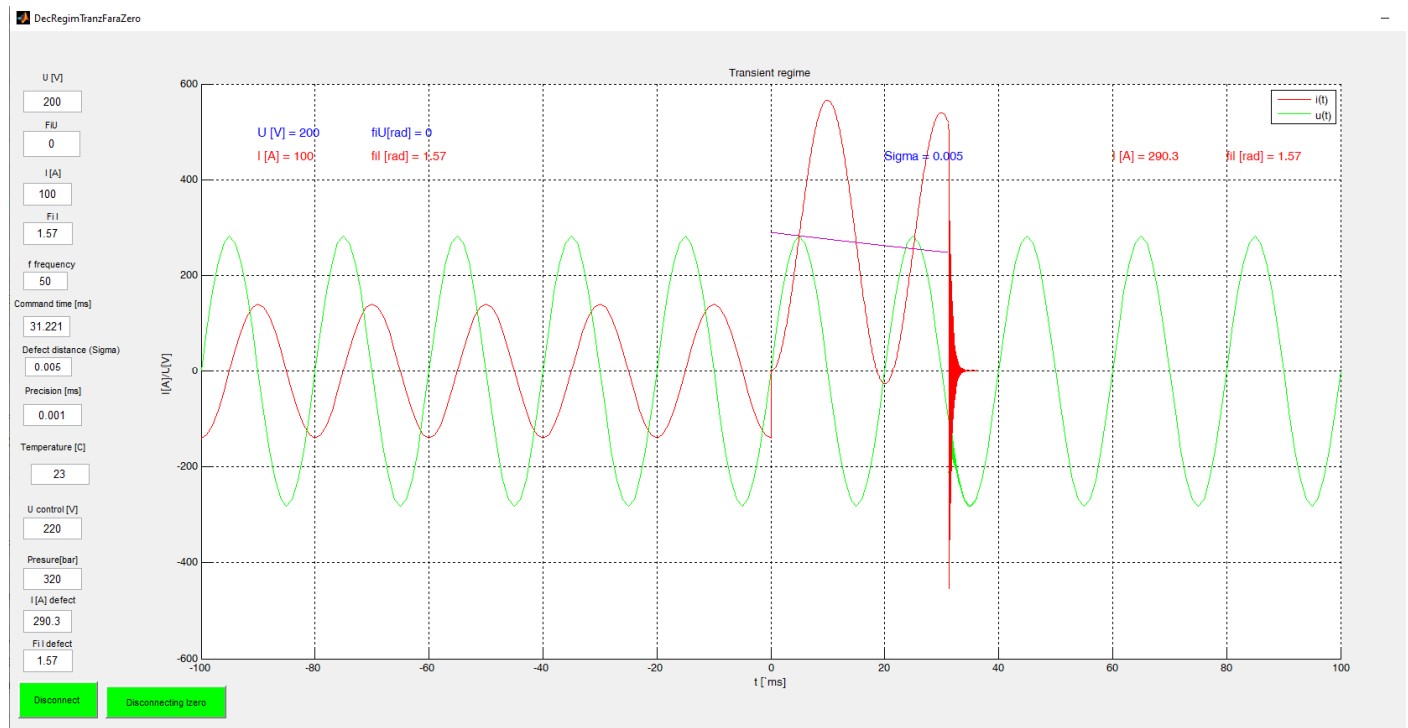

**Figure 16.** Transient uncontrolled switching diagram for inductive load, $\phi Idef = \pi/2$, Idef = 290.3 A, $\sigma = 0.005$, $t_{disconnect} = 31.221$ ms.

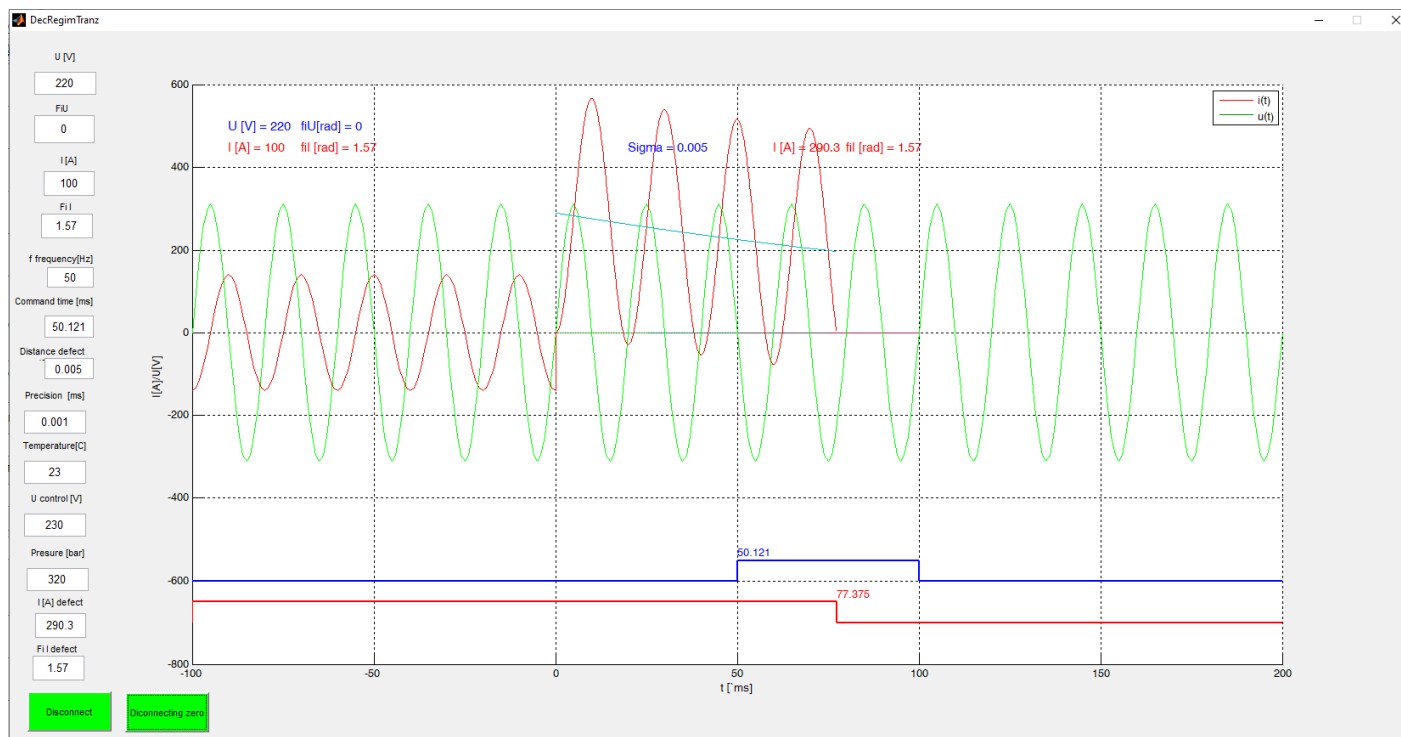

**Figure 17.** Transient switching diagram for inductive load, $\phi\text{Idef} = \pi/2$, Idef = 290.3 A, $\sigma$ = 0.005, $t_{command}$ = 50.121 ms, $t_{disconnect}$ = 77.375 ms.

The following values were used for the example in Figure 16: U = 220 V, $\varphi$U = 0 rad; I = 100 A, $\varphi$I = 1.57 = $\pi/2$ rad; f = 50 Hz, Sigma fault distance = 0.005; Idef = 290.3 A, $\varphi$ Idef = 1.57 = $\pi/2$ rad; command time = 31.221 ms.

This time is determined by the protection that detects the occurrence of the defect.

From Figure 16, it is found that the current to be interrupted is high and, within a few ms, it is damped, requiring a significant process to extinguish the electric arc. The fault current from Figure 16 must be interrupted very soon after its occurrence, because if it is maintained for a long time, it can lead to the destruction of the electrical equipment in operation.

To simulate the controlled disconnection shown in Figure 17, the following values were used: U = 220 V, $\varphi$U = 0 rad; I = 100 A, $\varphi$I = 1.57 = $\pi/2$ rad; f = 50 Hz, Sigma fault distance = 0.005; Idef = 290.3 A, $\varphi$ Idef = 1.57 = $\pi/2$ rad; command time = 50.121 ms.

This time is about half the amplitude of the fault current. It is observed that if the opening occurred at this time, we would have a damped variation of current and voltage [36]. The program determines the time when the fault current crosses the time axis at t = 97.108 ms and opens the switch so that the momentary value of the fault current is almost zero.

Figure 18 shows the disconnecting operation of the circuit breaker for a capacitive load.

At a command time greater than 150 ms, all three situations, capacitive, resistive, and inductive load, are depicted in Figure 19. It can be seen that in the case of a value of $\sigma$ = 0.005, at a command time over than 150 ms, the drive time is shorter than in the previous example, indifferent to the load, and is approximately 23 ms. The disconnect switch command date is held at 50 ms in the blue graphic representation, and the circuit breaker position is shown in red in the lower part. It is noted that the drive time between issuing the opening commands and disconnection of the circuit breaker is approximately 27 ÷ 28 ms in the case of a transient fault.

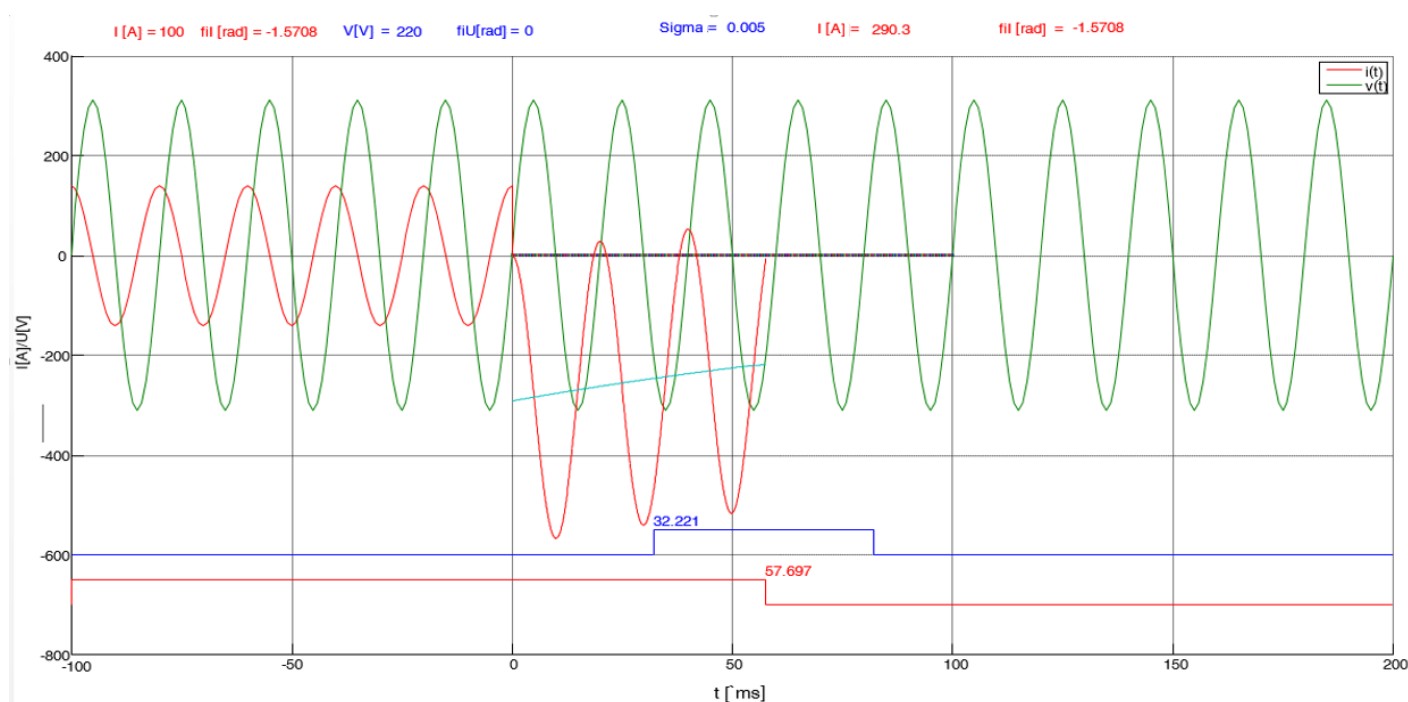

**Figure 18.** Transient switching diagram for capacitive load, $\phi I_{def} = \pi/2$, $I_{def} = 290.3$ A, $\sigma = 0.005$, $t_{command} = 32.221$ ms, $t_{disconnect} = 57.697$ ms.

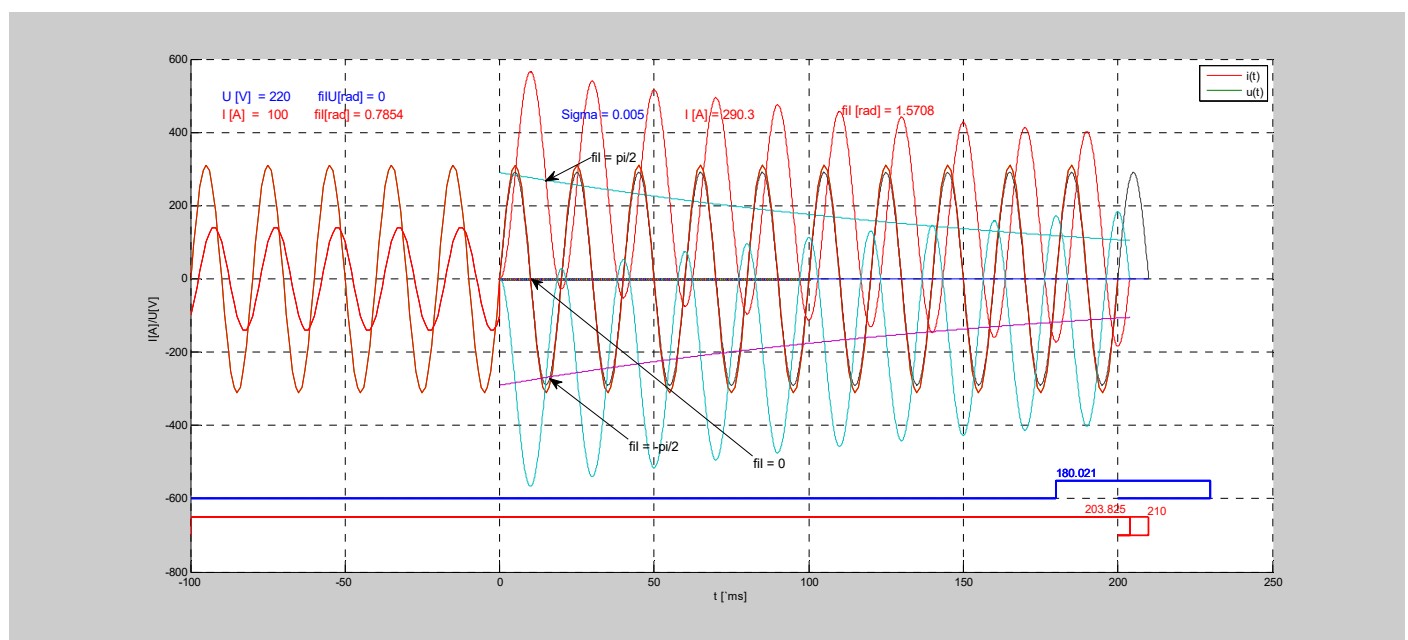

**Figure 19.** Transient switching diagram for capacitive, resistive and inductive load, $\phi I_{def} = -\pi/2$; 0; $\pi/2$, for $I_{def} = 290.3$ A, $\sigma = 0.005$, $t_{command} = 180.021$ ms, $t_{dconnect} = 203$ ms.

### 4.1.2. Simulation in Matlab of the Transient Regime of the SF6 Circuit-Breaker Connection

In the scope of studying the connection of a circuit breaker, the same scheme as in Figure 4 was used. The time coefficients on the connection process of the circuit-breaker are experimentally determined for the studied circuit-breakers and the results are presented in the following.

The temperature dependency of time coefficient $C_{temp} = f(T)$ on the connection of a circuit breaker is represented in Table 7. Taking into consideration the measured values, the equation which describes this dependency is presented in Equation (20).

$$C_{temp} (T) = 0.0026{\cdot}T^2 - 0.0829{\cdot}T + 71.482 \tag{20}$$

**Table 7.** Measured values of the dependency of time coefficient on outdoor temperature.

| T [°C] | $C_{temp}$ (ms) |
|---|---|
| 40 | 71.5 |
| 20 | 72 |
| 0 | 72.5 |
| −20 | 73 |
| −30 | 74 |
| −40 | 81 |

Figure 20 represents the dependency of parameter $C_{temp}$ on the outdoor temperature on the connection of a circuit breaker.

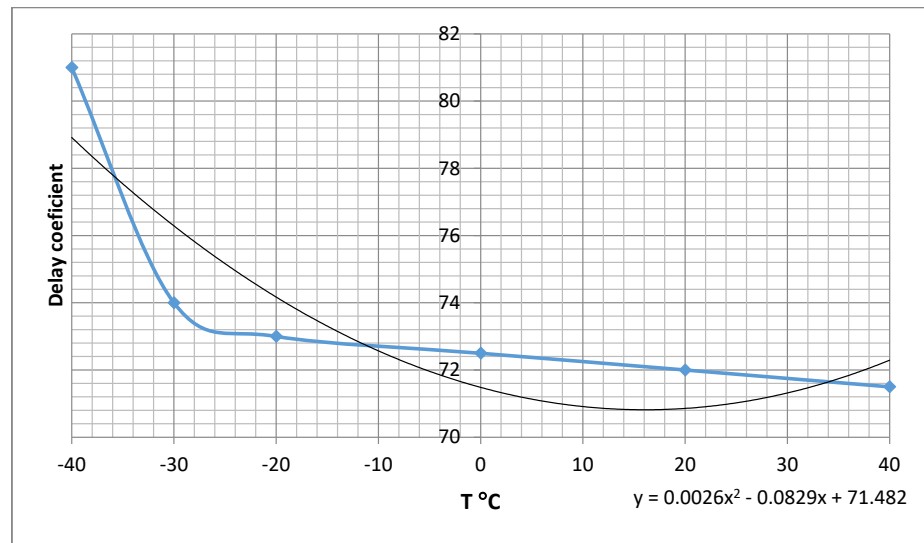

**Figure 20.** Dependency of **$C_{temp}$** on the outdoor temperature on the connection of a circuit breaker.

The DC voltage supply dependency of time coefficient $t_{UDC} = f (UDC)$ on the disconnection of a circuit breaker is represented in Table 8. Taking into consideration the measured values, the equation which describes the dependency is presented in Equation (21).

$$t_{UDC} (UDC) = -0.2{\cdot}UDC + 44 \tag{21}$$

**Table 8.** Measured values of the dependency of time coefficient on outdoor temperature.

| U Control (V) UDC | $t_{UDC}$ (ms) |
|---|---|
| 187 | 6.6 |
| 192 | 5.6 |
| 203 | 3.4 |
| 209 | 2.2 |
| 221 | −0.2 |
| 232 | −2.4 |
| 241 | −4.2 |
| 252 | −6.4 |
| 255 | −7 |

Figure 21 represents the dependency of parameter $t_{UDC}$ on the DC voltage supply on the connection of a circuit breaker.

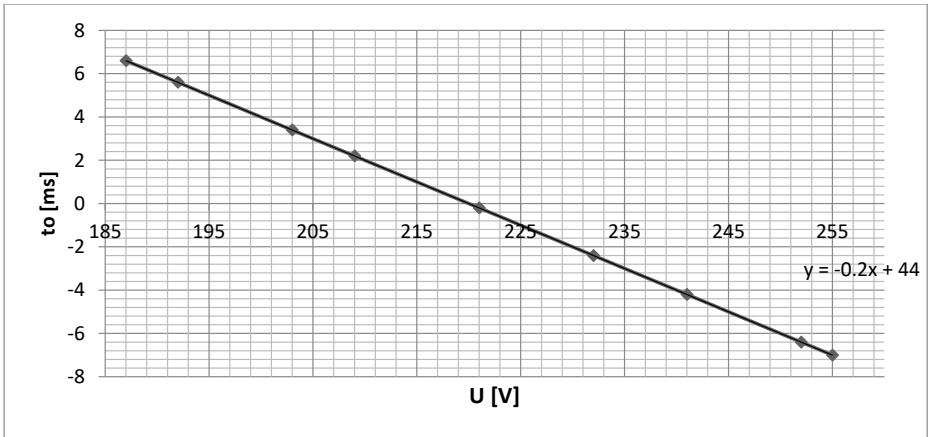

**Figure 21.** Dependency of $t_U$ on the DC voltage supply on the connection of a circuit breaker.

4.  The oil pressure dependency of time coefficient $C_{Ph} = f(P)$ on the connection of a circuit breaker is represented in Table 9. Taking into consideration the measured values, the equation which describes the dependency is presented in Equation (22).

**Table 9.** Measured values of the dependency of time coefficient on oil pressure.

| Oil Pressure (bar) | $C_{Ph}$ (ms) |
| --- | --- |
| 310 | 6 |
| 316 | 4.2 |
| 321 | 2.7 |
| 326 | 1.2 |
| 333 | −0.9 |
| 337 | −2.1 |
| 342 | −3.6 |
| 346 | −4.8 |
| 350 | −6 |

$$C_{Ph}\,(P) = -0.3 \cdot P + 99 \tag{22}$$

Figure 22 represents the dependency of parameter $C_{Ph}$ on the oil pressure on the connection of a circuit breaker.

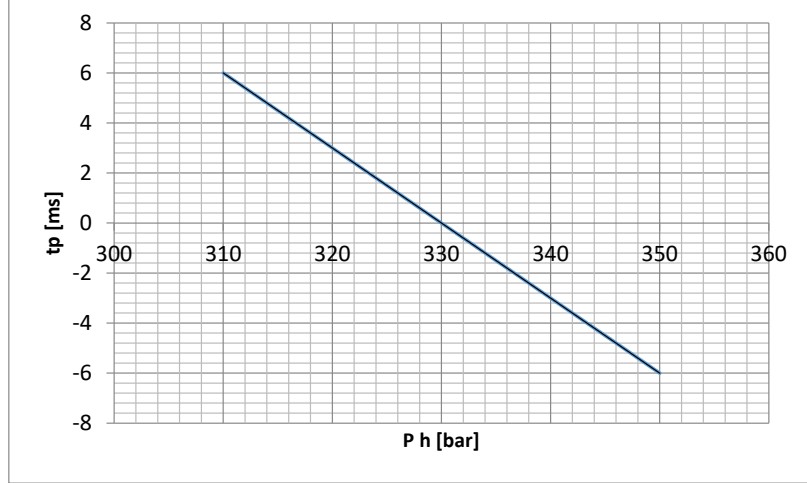

**Figure 22.** Time delay characteristic of the oil pressure on the connection of a circuit breaker.

For simulation of the connecting process, the following parameter values of a monophasic power supply system were used: U = 8.6 V, ɸU = −0.34°, f = 50 Hz, I = 3 A, ɸI = 1.26°, outdoor temperature = 18 °C, Ucc = 238 V, P = 325 bar.

Figure 23 presents the simulation result for the time command of connecting of circuit breaker of $t_1$ = $t_{command}$ = 5 ms.

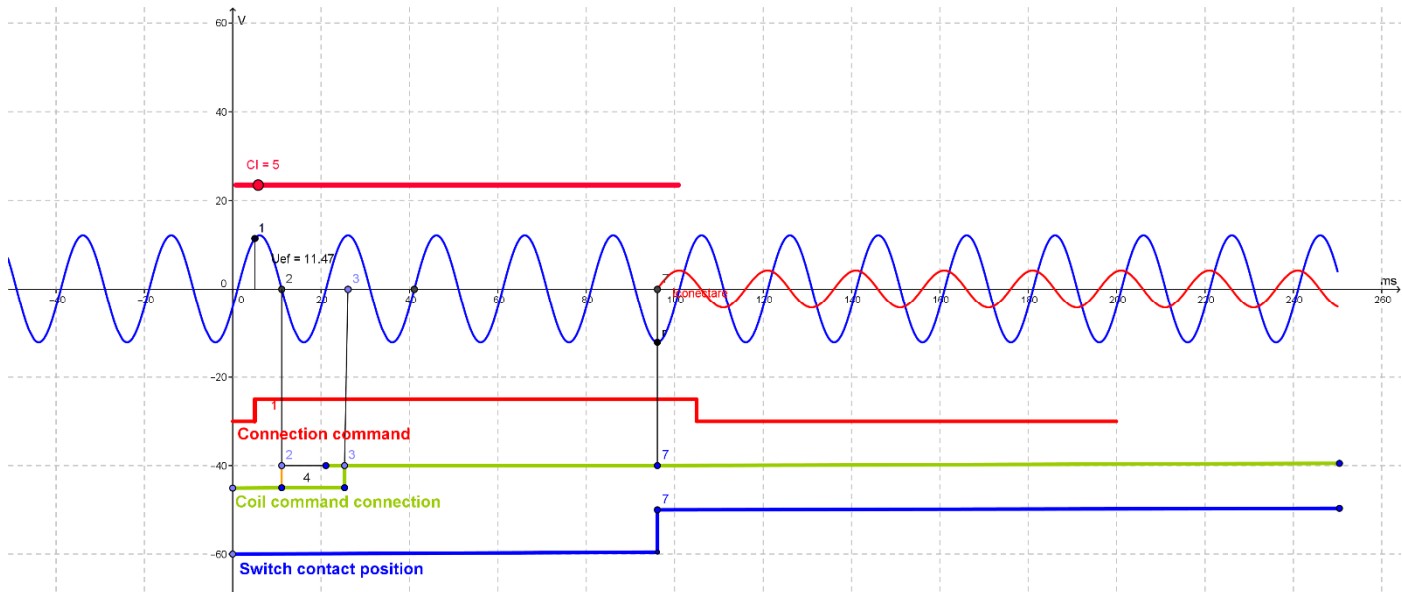

**Figure 23.** The simulation of monophasic power supply system on the connecting process of the circuit breaker on $t_1$ = $t_{command}$ = 5 ms.

The main parameters which are used in simulation of connection process of the circuit-breaker are presented in Table 10. Based on the used equation presented in Table 11, the parameter values of the connected simulation process are $t_1$ = $t_{command}$ = 5 ms.

**Table 10.** The main parameters used in simulation of connection process of the circuit-breaker.

| Parameters | Values |
|---|---|
| Temperature [°C] | 18 |
| Ucc [V] | 238 |
| P [bar] | 325 |
| tT | 71 |
| tucc | −4 |
| tp | 2 |
| t5 | 69 |
| t1 | 5 |
| t2 | 11 |

**Table 11.** The equation used in simulation of connection process of the circuit breaker.

| Equation | Parameters | Values |
|---|---|---|
| i(t) = 0 | t2 | 11 |
| t = f(T,U,P) | t5 | 69 |
| SP2 = 1000/f/3 | sp2 | 10 |
| nrp = t5/SP2 + 1 | nrp | 8 |
| t7 = t2 + nrp*SP2 | t7 | 96 |
| t4 = t7 − t6 | t4 | 27 |
| t3 = t4 − t2 | t3 | 38 |

### 5. Simulation Results, Industry Implementation and Related Discussion

*5.1. Simulation Results*

Controlled switching is used to eliminate transient regimes on manually switched capacitor banks, shunt reactors and power transformers. An important aspect of all controlled switching applications consists in the accuracy obtained during ignition and breaking of the electric arc.

Uncontrolled switching can lead to equipment wear. Closing resistors and coils that were previously used to reduce these problems are no longer required. Switching control provides an efficient solution. Taking into account the current values, the control unit optimizes the switching operations of the circuit breaker using instantaneous values of voltages or currents [28]. Disconnection of an inductive or capacitive load can have the effect of reigniting the electric arc and causing overvoltage. Connecting an inductive load can have the effect of overcurrent in the case of a non-optimized switching process.

The switching point at zero crossing of the current is a method of eliminating the transient effects that occur in time-controlled switching operations. Circuit breaker commands for connecting or disconnecting are delayed so that the contact is closed or opened at the optimum time characteristic of the phase angle. This method can improve the performance and life of the circuit breaker. Based on the simulations from the previous section, it can be seen, regarding the controlled switching of the transient regime, that the simulations allow us to obtain current and voltage wave shapes equivalent to those from the literature [11,37].

The simulations performed allowed the obtaining of different values of breaking currents, necessary in the disconnection process. In the literature, there are papers such as [38] that also investigate the breaking currents.

Based on simulation, it was possible to modify the character of the source, respectively, of the load by modifying the current and voltage phase shift. In this way, the types of interruptions could be analysed and compared with the circuit breaker tests presented in [39].

The simulation of the connecting time, in the case of circuit breakers with SF6, largely depends on the type of circuit-breaker. For each type of circuit breaker, the characteristics must be determined by the temperature, the DC supply voltage of the drive coils and the pressure of the hydraulic agent. Given that the circuit breaker operates under normal conditions, i.e., the temperature within $0\,°C \div +40\,°C$, we can approximate this coefficient as equal to 72 ms for the connection operation and 22 ms for the disconnection operation of the circuit breaker. The authors of [9] analyze the duration of the transient circulation current, defined as the time interval between the moment when this current appears and the moment when it disappears. This value was estimated at 12 ms and the simulations from this paper confirmed this.

In [8], research was conducted on the average interruption time, and it was proven that the values are 23–35 ms. To check these time intervals, it is necessary to use auxiliary contacts, which copy the position of the circuit breaker contacts, and provide very fast and accurate results [40].

As shown in Section 3, distance to fault was simulated in this paper using different values of the σ parameter, which influences the switching process, mainly the shape of the aperiodic component of the current and, thereby, the transient recovery voltage, as shown in [41]. Fault location is an important issue in the switching process. In other research, different approaches for fault location estimation were used [42].

*5.2. Industry Implementation*

This section describes the current implementation of the simulation results presented in the previous sections at a 400 kV power station. In this regard, the command-and-control system of a line circuit breaker currently in operation at the power station is presented.

Figure 24 depicts the control screen of a line cell. A line cell's command-and-control system must perform the following functions:

- The current monitoring function, which is carried out by displaying the station's single-phase operation scheme in dynamic coloring mode as well as the electrical

operating parameters of the equipment and installations that comprise the transformation station.

- The function of viewing information and detail screens, which can be achieved by accessing both the control screens and the dedicated surveillance screens for the existing units.
- The operator function, which includes connecting and disconnecting the user, transferring the remote control between levels of remote control and displaying the level at which it is, and inserting indicator boxes.

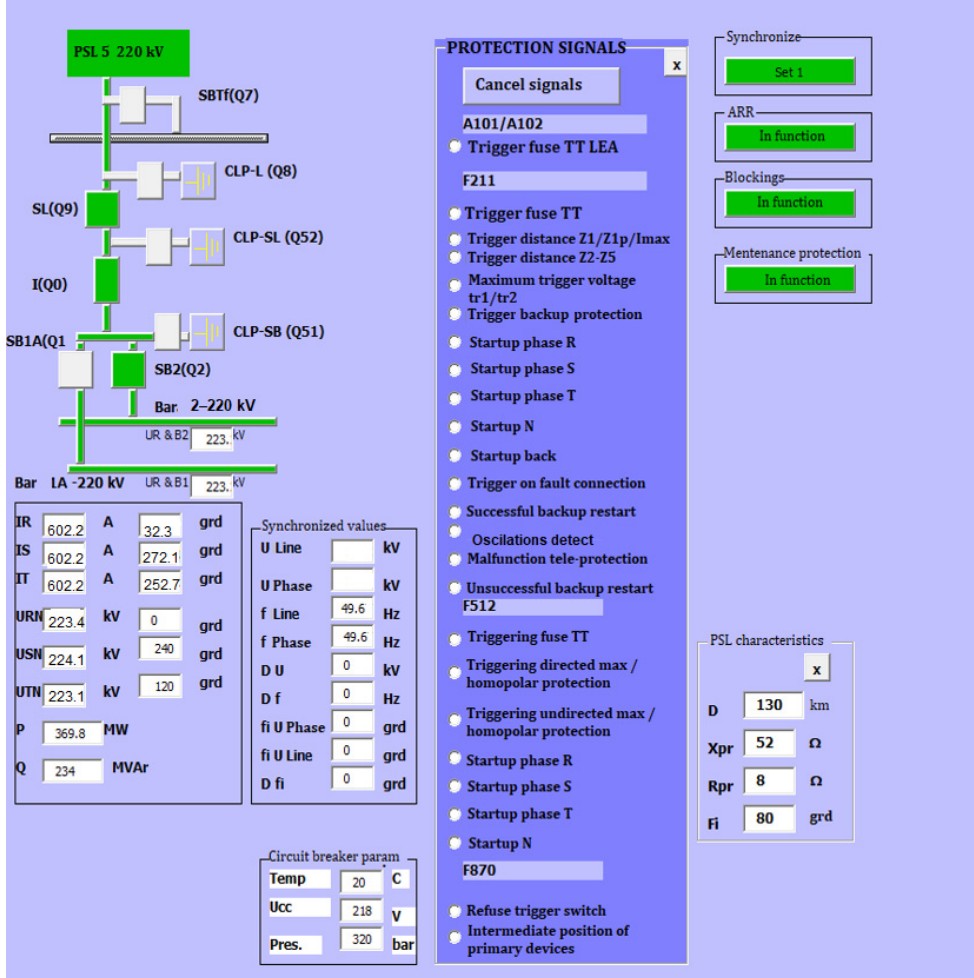

**Figure 24.** The control screen of a line cell.

The control screen displays information about the cell's electrical operating parameters (line voltage, current intensity on each phase, active power, reactive power and supply voltage frequency), as well as from the signaling system (information on the status of automations and protections and alarms from the cell protection system).

5.2.1. Controlled Disconnection of the Load Based on the Command-and-Control System, Using a Circuit Breaker

When the switch is pressed, a window with the following elements appears, as shown in Figure 25:

- a static label reading "Switch maneuver";
- a dynamic "Cancel" button for closing the window;
- two confirmation buttons for turning on the circuit breaker ("Disconnection" and "Controlled disconnection").

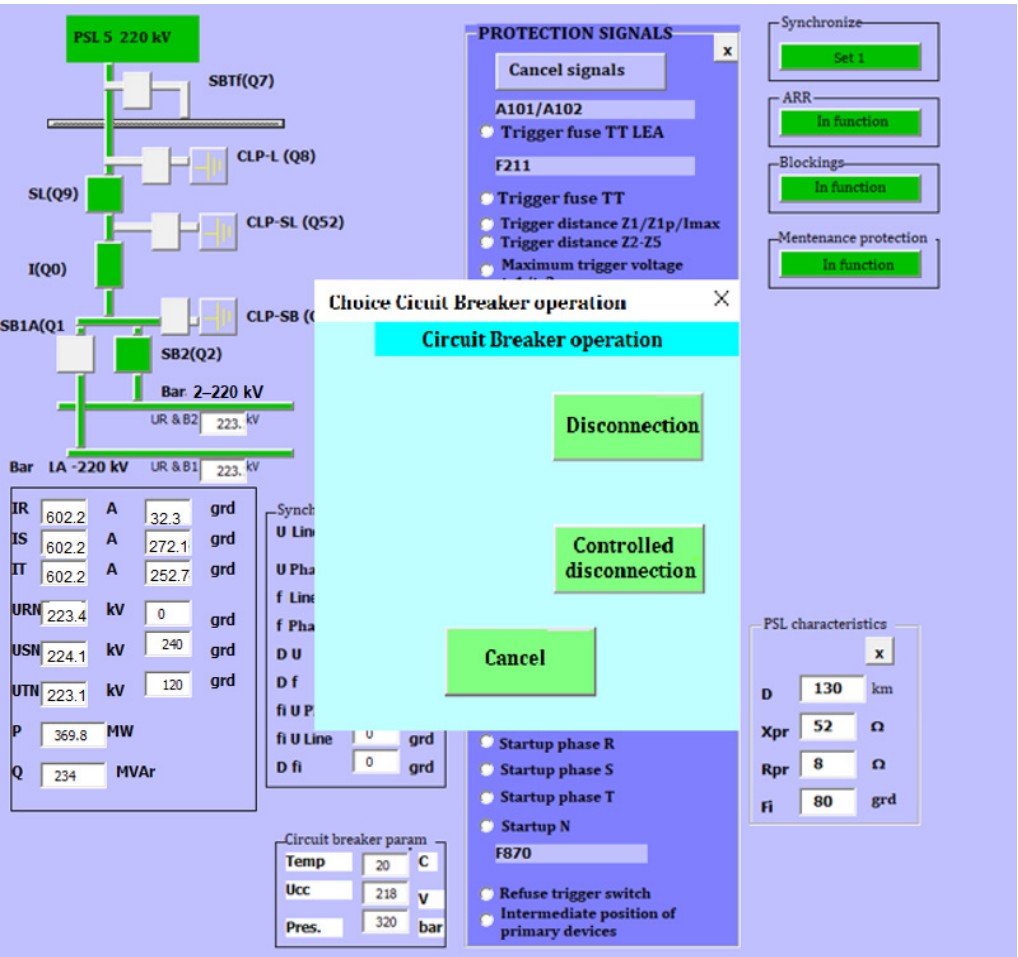

**Figure 25.** The control screen of the circuit breaker's disconnection.

When the "Disconnection" button is pressed, the circuit breaker is disconnected without any control.

If the "Controlled disconnection" button is selected, another window will open, as shown in Figure 26, with options to control the circuit breaker's disconnecting time, and containing the following elements:

- a static label with the word "Disconnect";
- a „Cancel" button for closing the window;
- a confirmation button labeled as "Save Data" that allows the circuit breaker to be opened and data to be saved from the moment of controlled opening.

5.2.2. Controlled Connection of the Load Based on the Command-and-Control System, Using a Circuit Breaker

When the switch is pressed, a window with the following elements appears, as shown in Figure 27:

- a static label with the words "Choice Circuit Breaker operation";
- a dynamic "Cancel" button for shutting down the window;
- two confirmation buttons for selecting the circuit breaker operation ("Synchronized connection" and "Non-synchronized connection");
- a dynamic "Controlled connection" button for using the controlled connection of the load;
- two confirmation buttons for activating the circuit breaker operation ("Activate Synchronizing" and "Un-activate synchronizing").

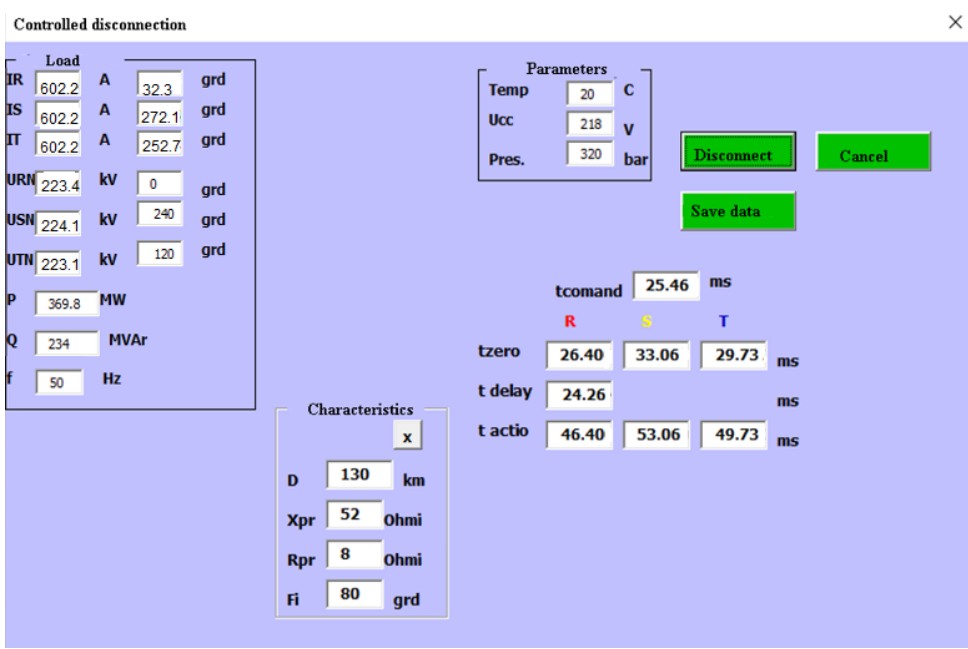

**Figure 26.** The control screen of the circuit breaker's controlled disconnection.

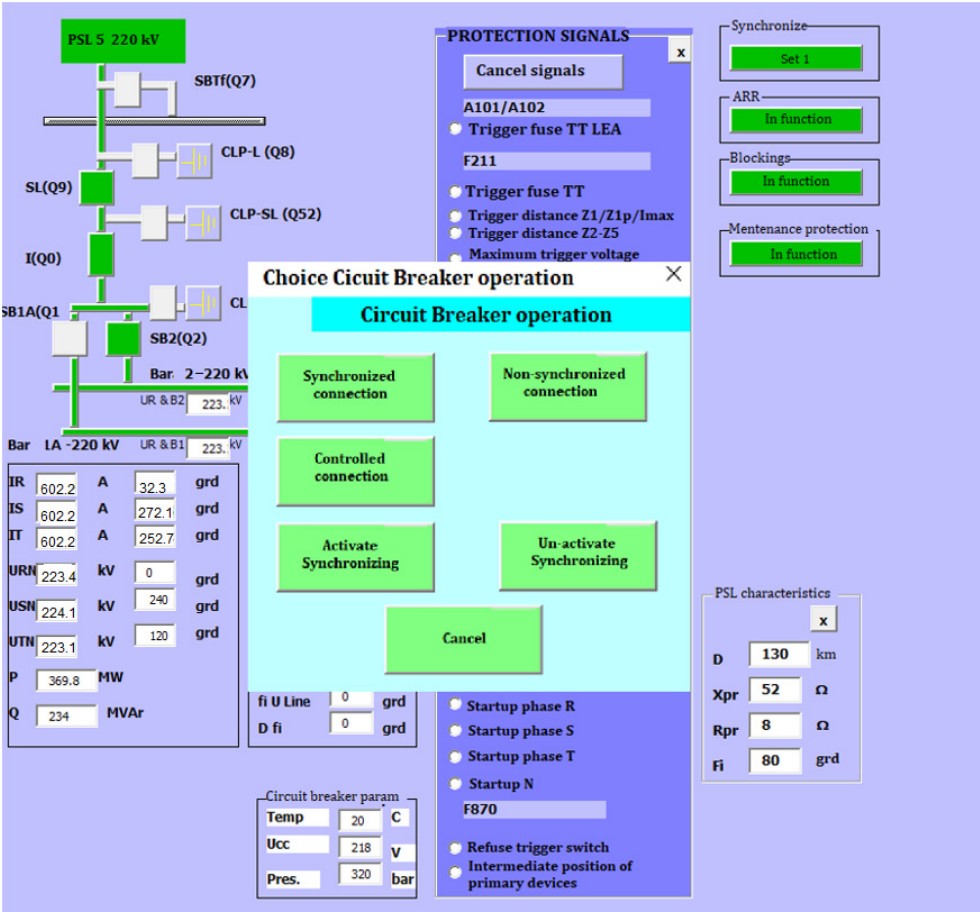

**Figure 27.** The control screen of the circuit breaker's connection.

If the "Controlled connection" button is selected, another window will open, as shown in Figure 28, with options to control the circuit breaker's connecting time, and containing the following elements:

- a static label with the word "Connection";
- a "Cancel" button for closing the window;
- a confirmation button labelled as "Save Data" that allows the circuit breaker to be opened and data to be saved from the moment of controlled opening.

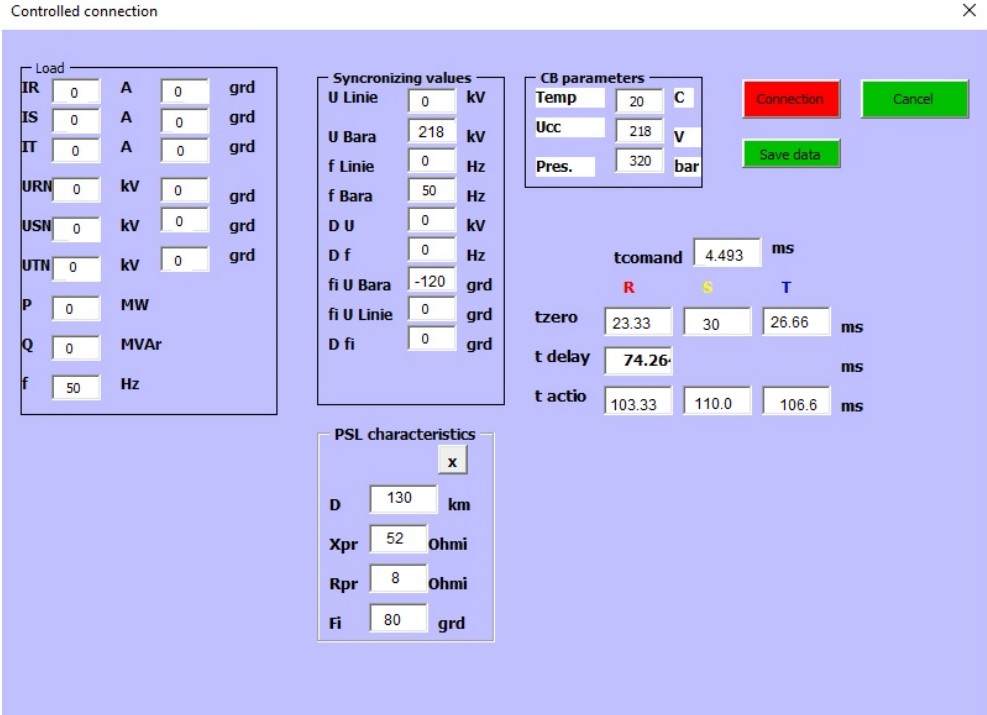

**Figure 28.** The control screen of the circuit breaker's controlled connection.

## 6. Conclusions

The controlled switching process has the advantage that, due to the small values of arcing times during the interruption of the fault, it can extend the lifespan of the circuit breaker.

Based on the presented simulation results, the following conclusions were identified: the calculation algorithm can be adapted according to the needs of the user since the control time can be set very close to zero, i.e., to the start moment of the transient regime. The main advantage of the program is that it calculates the moment of disconnection at zero crossing current. The defects caused by transient phenomena are influenced by the frequency of the intersection of the time axis, i.e., zero is not a constant factor due to the aperiodic component. The precision of the zero crossing point can be modified to the precision of 1 microsecond.

If it is not possible to determine the temperature, pressure and voltage characteristics, an approximation of the total disconnection time can be made from the field measurements and an error margin of 20 ms can be added.

Simulation results of the opening time for SF6 circuit breakers depend on the switch type. For each type of switch, its characteristics must be enhanced depending on the temperature, on the DC voltage supply and on the hydraulic agent pressure.

In situations in which the circuit breaker operates under normal conditions, i.e., the temperature within the range of 0 °C ÷ 40 °C, we can approximate the temperature coefficient to 22 ms for the opening operation. To check these times, it is necessary to use auxiliary contacts that copy the position of the circuit breaker contacts very quickly and accurately. The calculation algorithm for the connection time is simple. It all comes down to calculating the various time points and applying the command to the switch's coil at the appropriate time. Measurement is performed with high precision by means of existing transducers in the temperature and voltage.

The simulations were conducted for a single phase. If the circuit breaker control is monopolar, then the phase shift command must be transmitted in phase to the studied phase with a certain calculated delay.

Based on these observations, the system for implementing the theory presented in this paper in industry was presented in the second part of Section 5. This system was designed for the current protection system at an 800 MW power plant, specifically for the 400 kV high voltage line protection circuits. The protection system has been in use for nearly two years, and the results obtained thus far confirm that the research presented in the paper is fully applicable in industry.

**Author Contributions:** Conceptualization, C.P. and D.C.; methodology, C.P.; software, M.P. and D.C.; validation, S.M.; writing—original draft preparation, D.C. and S.M.; writing—review and editing, C.P. and M.P. All authors have read and agreed to the published version of the manuscript.

**Funding:** This research received no external funding.

**Institutional Review Board Statement:** Not applicable.

**Informed Consent Statement:** Not applicable.

**Conflicts of Interest:** The authors declare no conflict of interest.

## Nomenclature

| | |
|---|---|
| $R_p$ | parallel resistance |
| $C_p$ | parallel capacitance |
| $\Delta t$ | interruption time |
| $\Delta t_R$ | interruption time caused by resistance |
| $\Delta t_C$ | interruption time caused by capacitance |
| $i_L$ | inductivity current |
| $u_{arc}$ | arc voltage |
| $i_{arc}$ | arc current |
| $R_{arc}$ | arc resistance |
| $C_s$ | source capacitance |
| $C_l$ | load capacitance |
| $I_1$ | main switch |
| $I_2$ | auxiliary switch |
| Ro, Lo and Co | elements of oscillator branch |
| $v(t)$ | input voltage |
| $i(t)$ | input current |
| $\varphi v$ | voltage phase shift |
| $\varphi i$ | current phase shift |
| $f$ | frequency |
| $t_{delay}$ | delay time |
| $t_{breaker}$ | disconnected time |
| $t_{arc}$ | arc time |
| $t_{prearc}$ | prearc time |
| $i_{aper}$ | the aperiodic component of the current |
| $i_{per}$ | the periodic component of the current |
| $I_{def}$ | the amplitude of the current |
| $T$ | temperature |
| $P$ | oil pressure |
| $C_{temp}$ | temperature time coefficient |
| $t_{UDC}$ | DC voltage supply time coefficient |
| $C_{Ph}$ | oil pressure time coefficient |
| $t_{command}$ | connection/disconnection time command |

| $t_{disconnect}$ | disconnection time |
|---|---|
| HVCB | high voltage circuit breaker |
| GIS | gas insulated substations |
| TRV | transient recovery voltage |
| POW | point-on-wave |

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
