# Peer review of "Research Based on Modeling and Simulation of the Transient Regime in Controlled Switching with High Power Switches"

_machines, doi:10.3390/machines9050099_

Round 1
Reviewer 1 Report
The paper analyzes the controlled switching of high voltage circuit breakers, an important topic in electrical engineering. To this end, several types of switching transients were analyzed. This work is based on simulations including transient controlled connection/disconnection operations, analyzing different types of defects. The dependence of the switching process of a SF6 circuit breaker was analyzed based on different parameters, including the DC voltage supply, ambient temperature and oil pressure in the actuator.
However, the paper does not present experimental results, which are a good means of validating the developments made along the paper.
The Reviewer suggests a deep revision based on the following points:
- English grammar and style require a deep revision, otherwise the paper cannot be accepted.
- Equation (1) must be demonstrated or at least referred.
- Introduction section. There is not a clear link between figures 1 and 2. Please add detailed comments about the purpose of both figures.
- Equations (4) and (5) are not clear. Please explain better.
- There is no determinant in equation (7) as the authors claim.
- Values presented in Tables 1-11 must be referred.
- Equation (7) must be referred.
- Equation (19) is not clear.
- I suggest to include a nomenclature section including all parameters and variables dealt with along the paper.
- I highly recommend to include experimental results, at least those performed to determine the parameters presented in Tables 1-11.
- Reference list must be completely renewed and updated to include state of the art JCR journal references from 2019-2021.
I believe the remarks above would help the authors to improve the quality of the paper.
Author Response
Thank you for considering our manuscript for review and giving the opportunity for revision. We appreciate your time to process this manuscript and give review comments immediately. According to review comments and “Special Instructions to Authors”, we have made some revisions for this manuscript.

Reviewer 2 Report
With respect to the conference paper [ref. 8] presented by the authors to the 2020 8th 29 International Conference on Control, Mechatronics and Automation (ICCMA), in my opinion, it is necessary to further extend the contents of the submitted article, certainly by providing an accurate analysis of the scientific literature on the same issue in a new paragraph "Related works" currently absent.
Then, it is necessary to modify the article structure by adding a new Section 3 titled "Simulation results and related discussion" bringing in it some contents now in section 2 but also providing more results with respect to those already reported in the conference paper. Concerning the results presented in the sub-sections 2.1.1 and 2.1.2, the authors must provide more details and explanations on the reason for these results and not just report them merely in the various tables.
In addition, a comparison between the obtained simulation results and others presented in the literature and discussed in the section "Related work" (it has to be done) would be useful to increase the scientific level of the work.
Finally, the quality of most images needs to be improved.
Author Response

(The authors gave the same response as above.)

Round 2
Reviewer 1 Report
English grammar ans style require extensive revision
Author Response
- In comparison to the previous manuscript, the number of references has increased.
- English grammar and style were thoroughly revised, as evidenced by the attached file
- We believe that after grammar and style revisions, the results are more clearly presented.

Reviewer 2 Report
After adding new contents and increased the scientific strength in the review phase, in my opinion, the article can be accepted for publication in its current form.
Author Response
- The attached paper demonstrates that English grammar and style were thoroughly revised.
- We believe that the methods are better described after grammar and style revisions.
